# LAG: Lazily Aggregated Gradient for Communication-Efficient Distributed Learning

**Tianyi Chen**[*]     **Georgios B. Giannakis**[*]     **Tao Sun**[†,*]     **Wotao Yin**[*]

[*]University of Minnesota - Twin Cities, Minneapolis, MN 55455, USA
[†]National University of Defense Technology, Changsha, Hunan 410073, China
[*]University of California - Los Angeles, Los Angeles, CA 90095, USA
{chen3827,georgios@umn.edu}  nudtsuntao@163.com  wotaoyin@math.ucla.edu

## Abstract

This paper presents a new class of gradient methods for distributed machine learning that adaptively skip the gradient calculations to learn with reduced communication and computation. Simple rules are designed to detect slowly-varying gradients and, therefore, trigger the reuse of outdated gradients. The resultant gradient-based algorithms are termed **L**azily **A**ggregated **G**radient — justifying our acronym **LAG** used henceforth. Theoretically, the merits of this contribution are: i) the convergence rate is the same as batch gradient descent in strongly-convex, convex, and nonconvex cases; and, ii) if the distributed datasets are heterogeneous (quantified by certain measurable constants), the communication rounds needed to achieve a targeted accuracy are reduced thanks to the adaptive reuse of *lagged* gradients. Numerical experiments on both synthetic and real data corroborate a significant communication reduction compared to alternatives.

## 1 Introduction

In this paper, we develop communication-efficient algorithms to solve the following problem

$$\min_{\boldsymbol{\theta} \in \mathbb{R}^d} \mathcal{L}(\boldsymbol{\theta}) \quad \text{with} \quad \mathcal{L}(\boldsymbol{\theta}) := \sum_{m \in \mathcal{M}} \mathcal{L}_m(\boldsymbol{\theta}) \tag{1}$$

where $\boldsymbol{\theta} \in \mathbb{R}^d$ is the unknown vector, $\mathcal{L}$ and $\{\mathcal{L}_m, m \in \mathcal{M}\}$ are smooth (but not necessarily convex) functions with $\mathcal{M} := \{1, \ldots, M\}$. Problem (1) naturally arises in a number of areas, such as multi-agent optimization [1], distributed signal processing [2], and distributed machine learning [3]. Considering the distributed machine learning paradigm, each $\mathcal{L}_m$ is also a sum of functions, e.g., $\mathcal{L}_m(\boldsymbol{\theta}) := \sum_{n \in \mathcal{N}_m} \ell_n(\boldsymbol{\theta})$, where $\ell_n$ is the loss function (e.g., square or the logistic loss) with respect to the vector $\boldsymbol{\theta}$ (describing the model) evaluated at the training sample $\mathbf{x}_n$; that is, $\ell_n(\boldsymbol{\theta}) := \ell(\boldsymbol{\theta}; \mathbf{x}_n)$. While machine learning tasks are traditionally carried out at a single server, for datasets with massive samples $\{\mathbf{x}_n\}$, running gradient-based iterative algorithms at a single server can be prohibitively slow; e.g., the server needs to sequentially compute gradient components given limited processors. A simple yet popular solution in recent years is to parallelize the training across multiple computing units (a.k.a. workers) [3]. Specifically, assuming batch samples distributedly stored in a total of $M$ workers with the worker $m \in \mathcal{M}$ associated with samples $\{\mathbf{x}_n, n \in \mathcal{N}_m\}$, a globally shared model $\boldsymbol{\theta}$ will be updated at the central server by aggregating gradients computed by workers. Due to bandwidth and privacy concerns, each worker $m$ will not upload its data $\{\mathbf{x}_n, n \in \mathcal{N}_m\}$ to the server, thus the learning task needs to be performed by iteratively communicating with the server.

We are particularly interested in the scenarios where communication between the central server and the local workers is costly, as is the case with the Federated Learning setting [4, 5], the cloud-edge

AI systems [6], and more in the emerging Internet-of-Things paradigm [7]. In those cases, communication latency is the bottleneck of overall performance. More precisely, the communication latency is a result of initiating communication links, queueing and propagating the message. For sending small messages, e.g., the $d$-dimensional model $\boldsymbol{\theta}$ or aggregated gradient, this latency dominates the message size-dependent transmission latency. Therefore, it is important to reduce the number of communication rounds, even more so than the bits per round. In short, **our goal** is to find the model parameter $\boldsymbol{\theta}$ that minimizes (1) using as low communication overhead as possible.

## 1.1 Prior art

To put our work in context, we review prior contributions that we group in two categories.

**Large-scale machine learning.** Solving (1) at a single server has been extensively studied for large-scale learning tasks, where the "workhorse approach" is the simple yet efficient stochastic gradient descent (SGD) [8, 9]. Albeit its low per-iteration complexity, the inherited variance prevents SGD to achieve fast convergence. Recent advances include leveraging the so-termed *variance reduction* techniques to achieve both low complexity and fast convergence [10–12]. For learning beyond a single server, distributed parallel machine learning is an attractive solution to tackle large-scale learning tasks, where the parameter server architecture is the most commonly used one [3, 13]. Different from the single server case, parallel implementation of the batch gradient descent (GD) is a popular choice, since SGD that has low complexity per iteration requires a large number of iterations thus communication rounds [14]. For traditional parallel learning algorithms however, latency, bandwidth limits, and unexpected drain on resources, that delay the update of even a single worker will slow down the entire system operation. Recent research efforts in this line have been centered on understanding asynchronous-parallel algorithms to speed up machine learning by eliminating costly synchronization; e.g., [15–20]. All these approaches either reduce the *computational* complexity, or, reduce the *run time*, but they do not save communication.

**Communication-efficient learning.** Going beyond single-server learning, the high communication overhead becomes the bottleneck of the overall system performance [14]. Communication-efficient learning algorithms have gained popularity [21, 22]. Distributed learning approaches have been developed based on quantized (gradient) information, e.g., [23–26], but they only reduce the required bandwidth per communication, not the rounds. For machine learning tasks where the loss function is convex and its conjugate dual is expressible, the dual coordinate ascent-based approaches have been demonstrated to yield impressive empirical performance [5, 27, 28]. But these algorithms run in a double-loop manner, and the communication reduction has not been formally quantified. To reduce communication by accelerating convergence, approaches leveraging (inexact) second-order information have been studied in [29, 30]. Roughly speaking, algorithms in [5, 27–30] reduce communication by increasing local computation (relative to GD), while our method does not increase local computation. In settings *different* from the one considered in this paper, communication-efficient approaches have been recently studied with triggered communication protocols [31, 32]. Except for convergence guarantees however, no theoretical justification for communication reduction has been established in [31]. While a sublinear convergence rate can be achieved by algorithms in [32], the proposed gradient selection rule is nonadaptive and requires double-loop iterations.

## 1.2 Our contributions

Before introducing our approach, we revisit the popular GD method for (1) in the setting of one parameter server and $M$ workers: At iteration $k$, the server broadcasts the current model $\boldsymbol{\theta}^k$ to *all* the workers; every worker $m \in \mathcal{M}$ computes $\nabla\mathcal{L}_m(\boldsymbol{\theta}^k)$ and uploads it to the server; and once receiving gradients from all workers, the server updates the model parameters via

**GD iteration** $$\boldsymbol{\theta}^{k+1} = \boldsymbol{\theta}^k - \alpha\nabla_{\mathrm{GD}}^k \quad \text{with} \quad \nabla_{\mathrm{GD}}^k := \sum_{m \in \mathcal{M}} \nabla\mathcal{L}_m(\boldsymbol{\theta}^k) \tag{2}$$

where $\alpha$ is a stepsize, and $\nabla_{\mathrm{GD}}^k$ is an aggregated gradient that summarizes the model change. To implement (2), the server has to communicate with *all* workers to obtain fresh $\{\nabla\mathcal{L}_m(\boldsymbol{\theta}^k)\}$.

In this context, the present paper puts forward a new batch gradient method (as simple as GD) that can *skip* communication at certain rounds, which justifies the term **L**azily **A**ggregated **G**radient

| Metric | Communication | | Computation | | Memory | |
|---|---|---|---|---|---|---|
| Algorithm | PS→WK $m$ | WK $m$ →PS | PS | WK $m$ | PS | WK $m$ |
| GD | $\boldsymbol{\theta}^k$ | $\nabla\mathcal{L}_m$ | (2) | $\nabla\mathcal{L}_m$ | $\boldsymbol{\theta}^k$ | / |
| LAG-PS | $\boldsymbol{\theta}^k$, if $m\in\mathcal{M}^k$ | $\delta\nabla_m^k$, if $m\in\mathcal{M}^k$ | (4), (12b) | $\nabla\mathcal{L}_m$, if $m\in\mathcal{M}^k$ | $\boldsymbol{\theta}^k, \nabla^k, \{\hat{\boldsymbol{\theta}}_m^k\}$ | $\nabla\mathcal{L}_m(\hat{\boldsymbol{\theta}}_m^k)$ |
| LAG-WK | $\boldsymbol{\theta}^k$ | $\delta\nabla_m^k$, if $m\in\mathcal{M}^k$ | (4) | $\nabla\mathcal{L}_m$, (12a) | $\boldsymbol{\theta}^k, \nabla^k$ | $\nabla\mathcal{L}_m(\hat{\boldsymbol{\theta}}_m^k)$ |

Table 1: A comparison of communication, computation and memory requirements. **PS** denotes the parameter server, **WK** denotes the worker, **PS**→**WK** $m$ is the communication link from the server to the worker $m$, and **WK** $m \to$ **PS** is the communication link from the worker $m$ to the server.

(**LAG**). With its derivations deferred to Section 2, LAG resembles (2), given by

**LAG iteration**
$$\boldsymbol{\theta}^{k+1} = \boldsymbol{\theta}^k - \alpha\nabla^k \quad \text{with} \quad \nabla^k := \sum_{m\in\mathcal{M}} \nabla\mathcal{L}_m(\hat{\boldsymbol{\theta}}_m^k) \tag{3}$$

where each $\nabla\mathcal{L}_m(\hat{\boldsymbol{\theta}}_m^k)$ is either $\nabla\mathcal{L}_m(\boldsymbol{\theta}^k)$, when $\hat{\boldsymbol{\theta}}_m^k = \boldsymbol{\theta}^k$, or an outdated gradient that has been computed using an old copy $\hat{\boldsymbol{\theta}}_m^k \neq \boldsymbol{\theta}^k$. Instead of requesting fresh gradient from every worker in (2), the twist is to obtain $\nabla^k$ by refining the previous aggregated gradient $\nabla^{k-1}$; that is, using only the new gradients from the *selected* workers in $\mathcal{M}^k$, while reusing the outdated gradients from the rest of workers. Therefore, with $\hat{\boldsymbol{\theta}}_m^k := \boldsymbol{\theta}^k$, $\forall m\in\mathcal{M}^k$, $\hat{\boldsymbol{\theta}}_m^k := \hat{\boldsymbol{\theta}}_m^{k-1}$, $\forall m\notin\mathcal{M}^k$, LAG in (3) is equivalent to

**LAG iteration**
$$\boldsymbol{\theta}^{k+1} = \boldsymbol{\theta}^k - \alpha\nabla^k \quad \text{with} \quad \nabla^k = \nabla^{k-1} + \sum_{m\in\mathcal{M}^k} \delta\nabla_m^k \tag{4}$$

where $\delta\nabla_m^k := \nabla\mathcal{L}_m(\boldsymbol{\theta}^k) - \nabla\mathcal{L}_m(\hat{\boldsymbol{\theta}}_m^{k-1})$ is the difference between two evaluations of $\nabla\mathcal{L}_m$ at the current iterate $\boldsymbol{\theta}^k$ and the old copy $\hat{\boldsymbol{\theta}}_m^{k-1}$. If $\nabla^{k-1}$ is stored in the server, this simple modification scales down the per-iteration communication rounds from GD's $M$ to LAG's $|\mathcal{M}^k|$.

We develop two different rules to select $\mathcal{M}^k$. The first rule is adopted by the parameter server (PS), and the second one by every worker (WK). At iteration $k$,
**LAG-PS**: the server determines $\mathcal{M}^k$ and sends $\boldsymbol{\theta}^k$ to the workers in $\mathcal{M}^k$; each worker $m\in\mathcal{M}^k$ computes $\nabla\mathcal{L}_m(\boldsymbol{\theta}^k)$ and uploads $\delta\nabla_m^k$; each worker $m\notin\mathcal{M}^k$ does nothing; the server updates via (4);
**LAG-WK**: the server broadcasts $\boldsymbol{\theta}^k$ to all workers; every worker computes $\nabla\mathcal{L}_m(\boldsymbol{\theta}^k)$, and checks if it belongs to $\mathcal{M}^k$; only the workers in $\mathcal{M}^k$ upload $\delta\nabla_m^k$; the server updates via (4).
See a comparison of two LAG variants with GD in Table 1.

Naively reusing outdated gradients, while saving communication per iteration, can increase the total number of iterations. To keep this number in control, we judiciously design our simple trigger rules so that LAG can: i) achieve the *same* order of convergence rates (thus iteration complexities) as batch GD under strongly-convex, convex, and nonconvex smooth cases; and, ii) require *reduced* communication to achieve a targeted learning accuracy, when the distributed datasets are heterogeneous (measured by certain quantity specified later). In certain learning settings, LAG requires only $\mathcal{O}(1/M)$ communication of GD. Empirically, we found that LAG can reduce the communication required by GD and other distributed learning methods by an order of magnitude.

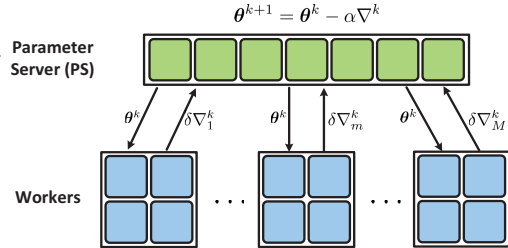

Figure 1: LAG in a parameter server setup.

**Notation**. Bold lowercase letters denote column vectors, which are transposed by $(\cdot)^\top$. And $\|\mathbf{x}\|$ denotes the $\ell_2$-norm of $\mathbf{x}$. Inequalities for vectors $\mathbf{x} > \mathbf{0}$ is defined entrywise.

## 2  LAG: Lazily Aggregated Gradient Approach

In this section, we formally develop our LAG method, and present the intuition and basic principles behind its design. The original idea of LAG comes from a simple rewriting of the GD iteration (2) as

$$\boldsymbol{\theta}^{k+1} = \boldsymbol{\theta}^k - \alpha \sum_{m\in\mathcal{M}} \nabla\mathcal{L}_m(\boldsymbol{\theta}^{k-1}) - \alpha \sum_{m\in\mathcal{M}} \left(\nabla\mathcal{L}_m(\boldsymbol{\theta}^k) - \nabla\mathcal{L}_m(\boldsymbol{\theta}^{k-1})\right). \tag{5}$$

Let us view $\nabla\mathcal{L}_m(\boldsymbol{\theta}^k) - \nabla\mathcal{L}_m(\boldsymbol{\theta}^{k-1})$ as a refinement to $\nabla\mathcal{L}_m(\boldsymbol{\theta}^{k-1})$, and recall that obtaining this refinement requires a round of communication between the server and the worker $m$. Therefore, to save communication, we can skip the server's communication with the worker $m$ if this refinement is small compared to the old gradient; that is, $\|\nabla\mathcal{L}_m(\boldsymbol{\theta}^k) - \nabla\mathcal{L}_m(\boldsymbol{\theta}^{k-1})\| \ll \|\sum_{m\in\mathcal{M}}\nabla\mathcal{L}_m(\boldsymbol{\theta}^{k-1})\|$.

Generalizing on this intuition, given the generic outdated gradient components $\{\nabla\mathcal{L}_m(\hat{\boldsymbol{\theta}}_m^{k-1})\}$ with $\hat{\boldsymbol{\theta}}_m^{k-1} = \boldsymbol{\theta}_m^{k-1-\tau_m^{k-1}}$ for a certain $\tau_m^{k-1} \geq 0$, if communicating with some workers will bring only small gradient refinements, we skip those communications (contained in set $\mathcal{M}_c^k$) and end up with

$$\boldsymbol{\theta}^{k+1} = \boldsymbol{\theta}^k - \alpha \sum_{m\in\mathcal{M}} \nabla\mathcal{L}_m(\hat{\boldsymbol{\theta}}_m^{k-1}) - \alpha \sum_{m\in\mathcal{M}^k} \left(\nabla\mathcal{L}_m(\boldsymbol{\theta}^k) - \nabla\mathcal{L}_m(\hat{\boldsymbol{\theta}}_m^{k-1})\right) \tag{6a}$$

$$= \boldsymbol{\theta}^k - \alpha\nabla\mathcal{L}(\boldsymbol{\theta}^k) - \alpha \sum_{m\in\mathcal{M}_c^k} \left(\nabla\mathcal{L}_m(\hat{\boldsymbol{\theta}}_m^{k-1}) - \nabla\mathcal{L}_m(\boldsymbol{\theta}^k)\right) \tag{6b}$$

where $\mathcal{M}^k$ and $\mathcal{M}_c^k$ are the sets of workers that *do* and *do not* communicate with the server, respectively. It is easy to verify that (6) is identical to (3) and (4). Comparing (2) with (6b), when $\mathcal{M}_c^k$ includes more workers, more communication is saved, but $\boldsymbol{\theta}^k$ is updated by a coarser gradient.

Key to addressing this communication versus accuracy tradeoff is a principled criterion to select a subset of workers $\mathcal{M}_c^k$ that do not communicate with the server at each round. To achieve this "sweet spot," we will rely on the fundamental descent lemma. For GD, it is given as follows [33].

**Lemma 1 (GD descent in objective)** *Suppose $\mathcal{L}(\boldsymbol{\theta})$ is L-smooth, and $\bar{\boldsymbol{\theta}}^{k+1}$ is generated by running one-step GD iteration* (2) *given $\boldsymbol{\theta}^k$ and stepsize $\alpha$. Then the objective values satisfy*

$$\mathcal{L}(\bar{\boldsymbol{\theta}}^{k+1}) - \mathcal{L}(\boldsymbol{\theta}^k) \leq -\left(\alpha - \frac{\alpha^2 L}{2}\right)\|\nabla\mathcal{L}(\boldsymbol{\theta}^k)\|^2 := \Delta_{\text{GD}}^k(\boldsymbol{\theta}^k). \tag{7}$$

Likewise, for our wanted iteration (6), the following holds; its proof is given in the Supplement.

**Lemma 2 (LAG descent in objective)** *Suppose $\mathcal{L}(\boldsymbol{\theta})$ is L-smooth, and $\boldsymbol{\theta}^{k+1}$ is generated by running one-step LAG iteration* (4) *given $\boldsymbol{\theta}^k$. Then the objective values satisfy (cf. $\delta\nabla_m^k$ in* (4))

$$\mathcal{L}(\boldsymbol{\theta}^{k+1}) - \mathcal{L}(\boldsymbol{\theta}^k) \leq -\frac{\alpha}{2}\left\|\nabla\mathcal{L}(\boldsymbol{\theta}^k)\right\|^2 + \frac{\alpha}{2}\left\|\sum_{m\in\mathcal{M}_c^k}\delta\nabla_m^k\right\|^2 + \left(\frac{L}{2} - \frac{1}{2\alpha}\right)\left\|\boldsymbol{\theta}^{k+1} - \boldsymbol{\theta}^k\right\|^2 := \Delta_{\text{LAG}}^k(\boldsymbol{\theta}^k). \tag{8}$$

Lemmas 1 and 2 estimate the objective value descent by performing one-iteration of the GD and LAG methods, respectively, conditioned on a common iterate $\boldsymbol{\theta}^k$. GD finds $\Delta_{\text{GD}}^k(\boldsymbol{\theta}^k)$ by performing $M$ rounds of communication with all the workers, while LAG yields $\Delta_{\text{LAG}}^k(\boldsymbol{\theta}^k)$ by performing only $|\mathcal{M}^k|$ rounds of communication with a selected subset of workers. Our pursuit is to select $\mathcal{M}^k$ to ensure that *LAG enjoys larger per-communication descent than GD*; that is

$$\Delta_{\text{LAG}}^k(\boldsymbol{\theta}^k)/|\mathcal{M}^k| \leq \Delta_{\text{GD}}^k(\boldsymbol{\theta}^k)/M. \tag{9}$$

Choosing the standard $\alpha = 1/L$, we can show that in order to guarantee (9), it is sufficient to have (see the supplemental material for the deduction)

$$\left\|\nabla\mathcal{L}_m(\hat{\boldsymbol{\theta}}_m^{k-1}) - \nabla\mathcal{L}_m(\boldsymbol{\theta}^k)\right\|^2 \leq \left\|\nabla\mathcal{L}(\boldsymbol{\theta}^k)\right\|^2/M^2, \quad \forall m\in\mathcal{M}_c^k. \tag{10}$$

However, directly checking (10) at each worker is expensive since obtaining $\|\nabla\mathcal{L}(\boldsymbol{\theta}^k)\|^2$ requires information from all the workers. Instead, we approximate $\|\nabla\mathcal{L}(\boldsymbol{\theta}^k)\|^2$ in (10) by

$$\left\|\nabla\mathcal{L}(\boldsymbol{\theta}^k)\right\|^2 \approx \frac{1}{\alpha^2}\sum_{d=1}^{D}\xi_d\left\|\boldsymbol{\theta}^{k+1-d} - \boldsymbol{\theta}^{k-d}\right\|^2 \tag{11}$$

where $\{\xi_d\}_{d=1}^{D}$ are constant weights, and the constant $D$ determines the number of recent iterate changes that LAG incorporates to approximate the current gradient. The rationale here is that, as $\mathcal{L}$ is smooth, $\nabla\mathcal{L}(\boldsymbol{\theta}^k)$ cannot be very different from the recent gradients or the recent iterate *lags*.

Building upon (10) and (11), we will include worker $m$ in $\mathcal{M}_c^k$ of (6) if it satisfies

**LAG-WK condition** $\qquad \left\|\nabla\mathcal{L}_m(\hat{\boldsymbol{\theta}}_m^{k-1}) - \nabla\mathcal{L}_m(\boldsymbol{\theta}^k)\right\|^2 \leq \frac{1}{\alpha^2 M^2}\sum_{d=1}^{D}\xi_d\left\|\boldsymbol{\theta}^{k+1-d} - \boldsymbol{\theta}^{k-d}\right\|^2. \tag{12a}$

| **Algorithm 1** LAG-WK | **Algorithm 2** LAG-PS |
|---|---|
| 1: **Input:** Stepsize $\alpha > 0$, and threshold $\{\xi_d\}$. | 1: **Input:** Stepsize $\alpha > 0$, $\{\xi_d\}$, and $L_m, \forall m$. |
| 2: **Initialize:** $\boldsymbol{\theta}^1, \{\nabla \mathcal{L}_m(\hat{\boldsymbol{\theta}}_m^0), \forall m\}$. | 2: **Initialize:** $\boldsymbol{\theta}^1, \{\hat{\boldsymbol{\theta}}_m^0, \nabla \mathcal{L}_m(\hat{\boldsymbol{\theta}}_m^0), \forall m\}$. |
| 3: **for** $k = 1, 2, \ldots, K$ **do** | 3: **for** $k = 1, 2, \ldots, K$ **do** |
| 4:     Server **broadcasts** $\boldsymbol{\theta}^k$ to all workers. | 4:     **for** worker $m = 1, \ldots, M$ **do** |
| 5:     **for** worker $m = 1, \ldots, M$ **do** | 5:         Server **checks** condition (12b). |
| 6:         Worker $m$ **computes** $\nabla \mathcal{L}_m(\boldsymbol{\theta}^k)$. | 6:         **if** worker $m$ violates (12b) **then** |
| 7:         Worker $m$ **checks** condition (12a). | 7:             Server **sends** $\boldsymbol{\theta}^k$ to worker $m$. |
| 8:         **if** worker $m$ violates (12a) **then** | 8:                 ▷ Save $\hat{\boldsymbol{\theta}}_m^k = \boldsymbol{\theta}^k$ at server |
| 9:             Worker $m$ **uploads** $\delta \nabla_m^k$. | 9:             Worker $m$ **computes** $\nabla \mathcal{L}_m(\boldsymbol{\theta}^k)$. |
| 10:                 ▷ Save $\nabla \mathcal{L}_m(\hat{\boldsymbol{\theta}}_m^k) = \nabla \mathcal{L}_m(\boldsymbol{\theta}^k)$ | 10:             Worker $m$ **uploads** $\delta \nabla_m^k$. |
| 11:         **else** | 11:         **else** |
| 12:             Worker $m$ uploads nothing. | 12:             No actions at server and worker $m$. |
| 13:         **end if** | 13:         **end if** |
| 14:     **end for** | 14:     **end for** |
| 15:     Server **updates** via (4). | 15:     Server **updates** via (4). |
| 16: **end for** | 16: **end for** |

Table 2: A comparison of LAG-WK and LAG-PS.

Condition (12a) is checked at *the worker side* after each worker receives $\boldsymbol{\theta}^k$ from the server and computes its $\nabla \mathcal{L}_m(\boldsymbol{\theta}^k)$. If broadcasting is also costly, we can resort to the following *server side* rule:

$$\textbf{LAG-PS condition} \qquad L_m^2 \left\| \hat{\boldsymbol{\theta}}_m^{k-1} - \boldsymbol{\theta}^k \right\|^2 \le \frac{1}{\alpha^2 M^2} \sum_{d=1}^{D} \xi_d \left\| \boldsymbol{\theta}^{k+1-d} - \boldsymbol{\theta}^{k-d} \right\|^2 . \qquad (12b)$$

The values of $\{\xi_d\}$ and $D$ admit simple choices, e.g., $\xi_d = 1/D, \forall d$ with $D = 10$ used in simulations.

**LAG-WK vs LAG-PS**. To perform (12a), the server needs to broadcast the current model $\boldsymbol{\theta}^k$, and all the workers need to compute the gradient; while performing (12b), the server needs the estimated smoothness constant $L_m$ for all the local functions. On the other hand, as it will be shown in Section 3, (12a) and (12b) lead to the same worst-case convergence guarantees. In practice, however, the server-side condition is more conservative than the worker-side one at communication reduction, because the smoothness of $\mathcal{L}_m$ readily implies that satisfying (12b) will necessarily satisfy (12a), but not vice versa. Empirically, (12a) will lead to a larger $\mathcal{M}_c^k$ than that of (12b), and thus extra communication overhead will be saved. Hence, (12a) and (12b) can be chosen according to users' preferences. LAG-WK and LAG-PS are summarized as Algorithms 1 and 2.

Regarding our proposed LAG method, three remarks are in order.

**R1)** With recursive update of the lagged gradients in (4) and the lagged iterates in (12), implementing LAG is as simple as GD; see Table 1. Both empirically and theoretically, we will further demonstrate that using lagged gradients even reduces the overall delay by cutting down costly communication.

**R2)** Although both LAG and asynchronous-parallel algorithms in [15–20] leverage stale gradients, they are very different. LAG *actively* creates staleness, and by design, it reduces total *communication* despite the staleness. Asynchronous algorithms *passively* receives staleness, and increases total communication due to the staleness, but it saves *run time*.

**R3)** Compared with existing efforts for communication-efficient learning such as quantized gradient, Nesterov's acceleration, dual coordinate ascent and second-order methods, LAG is not orthogonal to all of them. Instead, LAG can be combined with these methods to develop even more powerful learning schemes. Extension to the proximal LAG is also possible to cover nonsmooth regularizers.

## 3   Iteration and communication complexity

In this section, we establish the convergence of LAG, under the following standard conditions.

**Assumption 1**: Loss function $\mathcal{L}_m(\boldsymbol{\theta})$ is $L_m$-smooth, and $\mathcal{L}(\boldsymbol{\theta})$ is L-smooth.

**Assumption 2**: $\mathcal{L}(\boldsymbol{\theta})$ is convex and coercive.     **Assumption 3**: $\mathcal{L}(\boldsymbol{\theta})$ is $\mu$-strongly convex.

The subsequent convergence analysis critically builds on the following **Lyapunov function**:

$$\mathbb{V}^k := \mathcal{L}(\boldsymbol{\theta}^k) - \mathcal{L}(\boldsymbol{\theta}^*) + \sum_{d=1}^{D} \beta_d \left\| \boldsymbol{\theta}^{k+1-d} - \boldsymbol{\theta}^{k-d} \right\|^2 \qquad (13)$$

where $\boldsymbol{\theta}^*$ is the minimizer of (1), and $\{\beta_d\}$ is a sequence of constants that will be determined later.

We will start with the sufficient descent of our $\mathbb{V}^k$ in (13).

**Lemma 3 (descent lemma)** *Under Assumption 1, if $\alpha$ and $\{\xi_d\}$ are chosen properly, there exist constants $c_0, \cdots, c_D \geq 0$ such that the Lyapunov function in (13) satisfies*

$$\mathbb{V}^{k+1} - \mathbb{V}^k \leq -c_0 \left\| \nabla \mathcal{L}(\boldsymbol{\theta}^k) \right\|^2 - \sum_{d=1}^{D} c_d \left\| \boldsymbol{\theta}^{k+1-d} - \boldsymbol{\theta}^{k-d} \right\|^2 \qquad (14)$$

*which implies the descent in our Lyapunov function, that is, $\mathbb{V}^{k+1} \leq \mathbb{V}^k$.*

Lemma 3 is a generalization of GD's descent lemma. As specified in the supplementary material, under properly chosen $\{\xi_d\}$, the stepsize $\alpha \in (0, 2/L)$ including $\alpha = 1/L$ guarantees (14), matching the stepsize region of GD. With $\mathcal{M}^k = \mathcal{M}$ and $\beta_d = 0$, $\forall d$ in (13), Lemma 3 reduces to Lemma 1.

## 3.1 Convergence in strongly convex case

We first present the convergence under the smooth and strongly convex condition.

**Theorem 1 (strongly convex case)** *Under Assumptions 1-3, the iterates $\{\boldsymbol{\theta}^k\}$ of LAG satisfy*

$$\mathcal{L}(\boldsymbol{\theta}^K) - \mathcal{L}(\boldsymbol{\theta}^*) \leq \left(1 - c(\alpha; \{\xi_d\})\right)^K \mathbb{V}^0 \qquad (15)$$

*where $\boldsymbol{\theta}^*$ is the minimizer of $\mathcal{L}(\boldsymbol{\theta})$ in (1), and $c(\alpha; \{\xi_d\}) \in (0, 1)$ is a constant depending on $\alpha$, $\{\xi_d\}$ and $\{\beta_d\}$ and the condition number $\kappa := L/\mu$, which are specified in the supplementary material.*

**Iteration complexity**. The iteration complexity in its generic form is complicated since $c(\alpha; \{\xi_d\})$ depends on the choice of several parameters. Specifically, if we choose the parameters as follows

$$\xi_1 = \cdots = \xi_D := \xi < \frac{1}{D} \quad \text{and} \quad \alpha := \frac{1 - \sqrt{D\xi}}{L} \quad \text{and} \quad \beta_1 = \cdots = \beta_D := \frac{D-d+1}{2\alpha\sqrt{D/\xi}} \qquad (16)$$

then, following Theorem 1, the iteration complexity of LAG in this case is

$$\mathbb{I}_{\text{LAG}}(\epsilon) = \frac{\kappa}{1 - \sqrt{D\xi}} \log\left(\epsilon^{-1}\right). \qquad (17)$$

The iteration complexity in (17) is on the same order of GD's iteration complexity $\kappa \log(\epsilon^{-1})$, but has a worse constant. This is the consequence of using a smaller stepsize in (16) (relative to $\alpha = 1/L$ in GD) to simplify the choice of other parameters. Empirically, LAG with $\alpha = 1/L$ can achieve almost the same empirical iteration complexity as GD; see Section 4. Building on the iteration complexity, we study next the *communication complexity* of LAG. In the setting of our interest, we define the communication complexity as the *total number of uploads* over all the workers needed to achieve accuracy $\epsilon$. While the accuracy refers to the objective optimality error in the strongly convex case, it is considered as the gradient norm in general (non)convex cases.

The power of LAG is best illustrated by numerical examples; see an example of LAG-WK in Figure 2. Clearly, workers with a small smoothness constant communicate with the server less frequently. This intuition will be formally treated in the next lemma.

**Lemma 4 (lazy communication)** *Define the importance factor of every worker $m$ as $\mathbb{H}(m) := L_m/L$. If the stepsize $\alpha$ and the constants $\{\xi_d\}$ in the conditions (12) satisfy $\xi_D \leq \cdots \leq \xi_d \leq \cdots \leq \xi_1$ and worker $m$ satisfies*

$$\mathbb{H}^2(m) \leq \xi_d / (d\alpha^2 L^2 M^2) := \gamma_d \qquad (18)$$

*then, until the $k$-th iteration, worker $m$ communicates with the server at most $k/(d+1)$ rounds.*

Lemma 4 asserts that if the worker $m$ has a small $L_m$ (a close-to-linear loss function) such that $\mathbb{H}^2(m) \leq \gamma_d$, then under LAG, it only communicates with the server at most $k/(d+1)$ rounds. This is in contrast to the total of $k$ communication rounds involved per worker under GD. Ideally, we want as many workers satisfying (18) as possible, especially when $d$ is large.

To quantify the overall communication reduction, we define the **heterogeneity score function** as

$$h(\gamma) := \frac{1}{M} \sum_{m \in \mathcal{M}} \mathbb{1}(\mathbb{H}^2(m) \le \gamma) \qquad (19)$$

where the indicator $\mathbb{1}$ equals 1 when $\mathbb{H}^2(m) \le \gamma$ holds, and 0 otherwise. Clearly, $h(\gamma)$ is a nondecreasing function of $\gamma$, that depends on the distribution of smoothness constants $L_1, L_2, \ldots, L_M$. It is also instructive to view it as the cumulative distribution function of the *deterministic* quantity $\mathbb{H}^2(m)$, implying $h(\gamma) \in [0, 1]$. Putting it in our context, the critical quantity $h(\gamma_d)$ lower bounds the fraction of

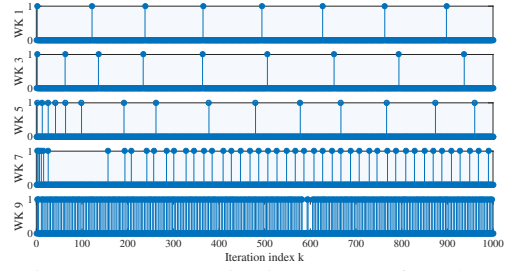

Figure 2: Communication events of workers $1, 3, 5, 7, 9$ over $1,000$ iterations. Each stick is an upload. A setup with $L_1 < \ldots < L_9$.

workers that communicate with the server at most $k/(d+1)$ rounds until the $k$-th iteration. We are now ready to present the communication complexity.

**Proposition 5 (communication complexity)** *With $\gamma_d$ defined in* (18) *and the function $h(\gamma)$ in* (19), *the communication complexity of LAG denoted as $\mathbb{C}_{\mathrm{LAG}}(\epsilon)$ is bounded by*

$$\mathbb{C}_{\mathrm{LAG}}(\epsilon) \le \left(1 - \sum_{d=1}^{D} \left(\frac{1}{d} - \frac{1}{d+1}\right) h(\gamma_d)\right) M \, \mathbb{I}_{\mathrm{LAG}}(\epsilon) := \left(1 - \Delta\bar{\mathbb{C}}(h; \{\gamma_d\})\right) M \, \mathbb{I}_{\mathrm{LAG}}(\epsilon) \qquad (20)$$

*where the constant is defined as $\Delta\bar{\mathbb{C}}(h; \{\gamma_d\}) := \sum_{d=1}^{D} \left(\frac{1}{d} - \frac{1}{d+1}\right) h(\gamma_d)$.*

The communication complexity in (20) crucially depends on the iteration complexity $\mathbb{I}_{\mathrm{LAG}}(\epsilon)$ as well as what we call the **fraction of reduced communication per iteration** $\Delta\bar{\mathbb{C}}(h; \{\gamma_d\})$. Simply choosing the parameters as (16), it follows from (17) and (20) that (cf. $\gamma_d = \xi(1 - \sqrt{D\xi})^{-2} M^{-2} d^{-1}$)

$$\mathbb{C}_{\mathrm{LAG}}(\epsilon) \le \left(1 - \Delta\bar{\mathbb{C}}(h; \xi)\right) \mathbb{C}_{\mathrm{GD}}(\epsilon) / \left(1 - \sqrt{D\xi}\right). \qquad (21)$$

where the GD's complexity is $\mathbb{C}_{\mathrm{GD}}(\epsilon) = M\kappa \log(\epsilon^{-1})$. In (21), due to the nondecreasing property of $h(\gamma)$, increasing the constant $\xi$ yields a smaller fraction of workers $1 - \Delta\bar{\mathbb{C}}(h; \xi)$ that are communicating per iteration, yet with a larger number of iterations (cf. (17)). The key enabler of LAG's communication reduction is a heterogeneous environment associated with a favorable $h(\gamma)$ ensuring that the benefit of increasing $\xi$ is more significant than its effect on increasing iteration complexity. More precisely, for a given $\xi$, if $h(\gamma)$ guarantees $\Delta\bar{\mathbb{C}}(h; \xi) > \sqrt{D\xi}$, then we have $\mathbb{C}_{\mathrm{LAG}}(\epsilon) < \mathbb{C}_{\mathrm{GD}}(\epsilon)$. Intuitively speaking, if there is a large fraction of workers with small $L_m$, LAG has lower communication complexity than GD. An example follows to illustrate this reduction.

**Example**. Consider $L_m = 1$, $m \ne M$, and $L_M = L \ge M^2 \gg 1$, where we have $\mathbb{H}(m) = 1/L, m \ne M$, $\mathbb{H}(M) = 1$, implying that $h(\gamma) \ge 1 - \frac{1}{M}$, if $\gamma \ge 1/L^2$. Choosing $D \ge M$ and $\xi = M^2 D/L^2 < 1/D$ in (16) such that $\gamma_D \ge 1/L^2$ in (18), we have (cf. (21))

$$\mathbb{C}_{\mathrm{LAG}}(\epsilon) / \mathbb{C}_{\mathrm{GD}}(\epsilon) \le \left[1 - \left(1 - \frac{1}{D+1}\right)\left(1 - \frac{1}{M}\right)\right] / \left(1 - MD/L\right) \approx \frac{M + D}{M(D+1)} \approx \frac{2}{M}. \qquad (22)$$

Due to technical issues in the convergence analysis, the current condition on $h(\gamma)$ to ensure LAG's communication reduction is relatively restrictive. Establishing communication reduction on a broader learning setting that matches the LAG's intriguing empirical performance is in our agenda.

## 3.2 Convergence in (non)convex case

LAG's convergence and communication reduction guarantees go beyond the strongly-convex case. We next establish the convergence of LAG for general convex functions.

**Theorem 2 (convex case)** *Under Assumptions 1 and 2, if $\alpha$ and $\{\xi_d\}$ are chosen properly, then*

$$\mathcal{L}(\boldsymbol{\theta}^K) - \mathcal{L}(\boldsymbol{\theta}^*) = \mathcal{O}\left(1/K\right). \qquad (23)$$

For nonconvex objective functions, LAG can guarantee the following convergence result.

**Theorem 3 (nonconvex case)** *Under Assumption 1, if $\alpha$ and $\{\xi_d\}$ are chosen properly, then*

$$\min_{1 \le k \le K} \left\|\boldsymbol{\theta}^{k+1} - \boldsymbol{\theta}^k\right\|^2 = o\left(1/K\right) \quad \text{and} \quad \min_{1 \le k \le K} \left\|\nabla\mathcal{L}(\boldsymbol{\theta}^k)\right\|^2 = o\left(1/K\right). \qquad (24)$$

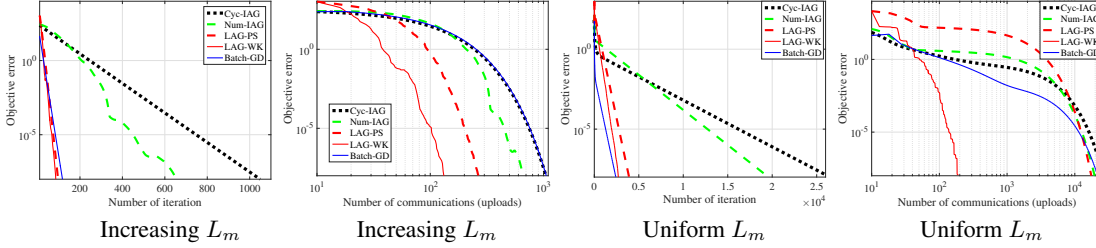

Figure 3: Iteration and communication complexity in synthetic datasets.

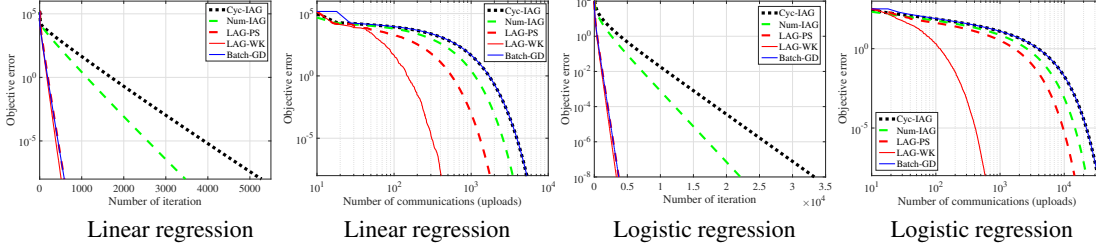

Figure 4: Iteration and communication complexity in real datasets.

Theorems 2 and 3 assert that with the judiciously designed lazy gradient aggregation rules, LAG can achieve order of convergence rate identical to GD for general (non)convex objective functions. Similar to Proposition 5, in the supplementary material, we have also shown that in the (non)convex case, LAG still requires less communication than GD, under certain conditions on the function $h(\gamma)$.

## 4   Numerical tests and conclusions

To validate the theoretical results, this section evaluates the empirical performance of LAG in linear and logistic regression tasks. All experiments were performed using MATLAB on an Intel CPU @ 3.4 GHz (32 GB RAM) desktop. By default, we consider one server, and nine workers. Throughout the test, we use $\mathcal{L}(\boldsymbol{\theta}^k) - \mathcal{L}(\boldsymbol{\theta}^*)$ as figure of merit of our solution. For logistic regression, the regularization parameter is set to $\lambda = 10^{-3}$. To benchmark LAG, we consider the following approaches.

▷ **Cyc-IAG** is the cyclic version of the incremental aggregated gradient (IAG) method [9, 10] that resembles the recursion (4), but communicates with one worker per iteration in a cyclic fashion.

▷ **Num-IAG** also resembles the recursion (4), and is the non-uniform-sampling enhancement of SAG [12], but it randomly selects one worker to obtain a fresh gradient per-iteration with the probability of choosing worker $m$ equal to $L_m / \sum_{m \in \mathcal{M}} L_m$.

▷ **Batch-GD** is the GD iteration (2) that communicates with all the workers per iteration.

For LAG-WK, we choose $\xi_d = \xi = 1/D$ with $D = 10$, and for LAG-PS, we choose more aggressive $\xi_d = \xi = 10/D$ with $D = 10$. Stepsizes for LAG-WK, LAG-PS, and GD are chosen as $\alpha = 1/L$; to optimize performance and guarantee stability, $\alpha = 1/(ML)$ is used in Cyc-IAG and Num-IAG.

We consider two **synthetic data** tests: a) linear regression with increasing smoothness constants, e.g., $L_m = (1.3^{m-1} + 1)^2$, $\forall m$; and, b) logistic regression with uniform smoothness constants, e.g., $L_1 = \ldots = L_9 = 4$; see Figure 3. For the case of increasing $L_m$, it is not surprising that both LAG variants need fewer communication rounds. Interesting enough, for uniform $L_m$, LAG-WK still has marked improvements on communication, thanks to its ability of exploiting the *hidden* smoothness of the loss functions; that is, the local curvature of $\mathcal{L}_m$ may not be as steep as $L_m$.

Performance is also tested on the **real datasets** [2]: a) linear regression using **Housing**, **Body fat**, **Abalone** datasets; and, b) logistic regression using **Ionosphere**, **Adult**, **Derm** datasets; see Figure 4. Each dataset is evenly split into three workers with the number of features used in the test equal to the minimal number of features among all datasets; see the details of parameters and data allocation in the supplement material. In all tests, LAG-WK outperforms the alternatives in terms of both metrics, especially reducing the needed communication rounds by several orders of magnitude. Its needed communication rounds can be even *smaller* than the number of iterations, if none of workers violate

|            | Linear regression |         |         | Logistic regression |         |         |
|------------|:-----------------:|:-------:|:-------:|:-------------------:|:-------:|:-------:|
| Algorithm  | $M=9$ | $M=18$ | $M=27$ | $M=9$ | $M=18$ | $M=27$ |
| Cyclic-IAG | 5271  | 10522  | 15773  | 33300 | 65287  | 97773  |
| Num-IAG    | 3466  | 5283   | 5815   | 22113 | 30540  | 37262  |
| **LAG-PS** | **1756** | **3610** | **5944** | **14423** | **29968** | **44598** |
| **LAG-WK** | **412**  | **657**  | **1058** | **584**   | **1098**  | **1723**  |
| Batch GD   | 5283  | 10548  | 15822  | 33309 | 65322  | 97821  |

Table 3: Communication complexity ($\epsilon = 10^{-8}$) in real dataset under different number of workers.

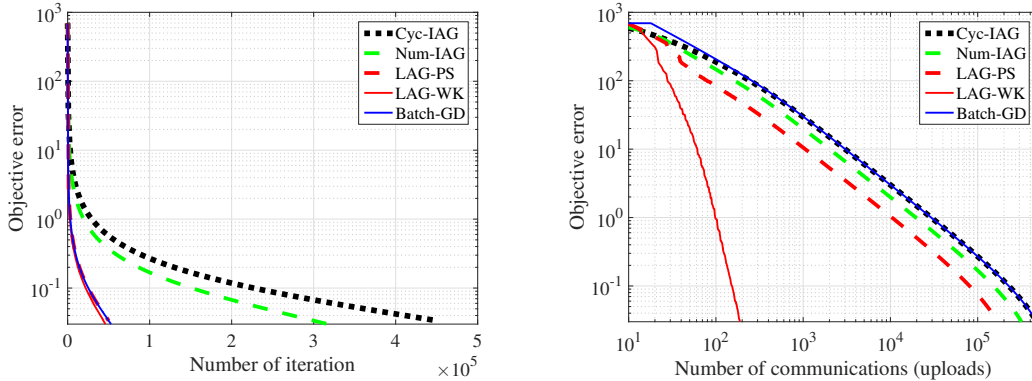

Figure 5: Iteration and communication complexity in Gisette dataset.

the trigger condition (12) at certain iterations. Additional tests under different number of workers are listed in Table 3, which corroborate the effectiveness of LAG when it comes to communication reduction. Similar performance gain has also been observed in the additional logistic regression test on a larger dataset **Gisette**. The dataset was taken from [7] which was constructed from the MNIST data [8]. After random selecting subset of samples and eliminating all-zero features, it contains 2000 samples $\mathbf{x}_n \in \mathbb{R}^{4837}$. We randomly split this dataset into nine workers. The performance of all the algorithms is reported in Figure 5 in terms of the iteration and communication complexity. Clearly, LAG-WK and LAG-PS achieve the same iteration complexity as GD, and outperform Cyc- and Num-IAG. Regarding communication complexity, two LAG variants reduce the needed communication rounds by several orders of magnitude compared with the alternatives.

Confirmed by the impressive empirical performance on both synthetic and real datasets, this paper developed a promising communication-cognizant method for distributed machine learning that we term Lazily Aggregated gradient (LAG) approach. LAG can achieve the same convergence rates as batch gradient descent (GD) in smooth strongly-convex, convex, and nonconvex cases, and requires fewer communication rounds than GD given that the datasets at different workers are heterogeneous. To overcome the limitations of LAG, future work consists of incorporating smoothing techniques to handle nonsmooth loss functions, and robustifying our aggregation rules to deal with cyber attacks.

### Acknowledgments

The work by T. Chen and G. Giannakis is supported in part by NSF 1500713 and 1711471, and NIH 1R01GM104975-01. The work by T. Chen is also supported by the Doctoral Dissertation Fellowship from the University of Minnesota. The work by T. Sun is supported in part by China Scholarship Council. The work by W. Yin is supported in part by NSF DMS-1720237 and ONR N0001417121.

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
