[Supplementary Material]

# Supplementary Document for "LAG: Lazily Aggregated Gradient for Communication-Efficient Distributed Learning"

In this supplementary document, we present the missing proofs of the lemmas and theorems in the main submission document.

## A   Proof of Lemma 2

Using the smoothness of $\mathcal{L}(\cdot)$ in Assumption 1, we have that

$$\mathcal{L}(\boldsymbol{\theta}^{k+1}) - \mathcal{L}(\boldsymbol{\theta}^k) \leq \left\langle \nabla\mathcal{L}(\boldsymbol{\theta}^k), \boldsymbol{\theta}^{k+1} - \boldsymbol{\theta}^k \right\rangle + \frac{L}{2} \left\| \boldsymbol{\theta}^{k+1} - \boldsymbol{\theta}^k \right\|^2. \tag{25}$$

Plugging (6) into $\left\langle \nabla\mathcal{L}(\boldsymbol{\theta}^k), \boldsymbol{\theta}^{k+1} - \boldsymbol{\theta}^k \right\rangle$ leads to (cf. $\hat{\boldsymbol{\theta}}_m^k = \hat{\boldsymbol{\theta}}_m^{k-1}, \forall m \in \mathcal{M}_c^k$)

$$\left\langle \nabla\mathcal{L}(\boldsymbol{\theta}^k), \boldsymbol{\theta}^{k+1} - \boldsymbol{\theta}^k \right\rangle$$

$$= -\alpha \left\langle \nabla\mathcal{L}(\boldsymbol{\theta}^k), \nabla\mathcal{L}(\boldsymbol{\theta}^k) + \sum_{m \in \mathcal{M}_c^k} \left( \nabla\mathcal{L}_m(\hat{\boldsymbol{\theta}}_m^k) - \nabla\mathcal{L}_m(\boldsymbol{\theta}^k) \right) \right\rangle$$

$$= -\alpha \left\| \nabla\mathcal{L}(\boldsymbol{\theta}^k) \right\|^2 - \alpha \left\langle \nabla\mathcal{L}(\boldsymbol{\theta}^k), \sum_{m \in \mathcal{M}_c^k} \left( \nabla\mathcal{L}_m(\hat{\boldsymbol{\theta}}_m^k) - \nabla\mathcal{L}_m(\boldsymbol{\theta}^k) \right) \right\rangle$$

$$= -\alpha \left\| \nabla\mathcal{L}(\boldsymbol{\theta}^k) \right\|^2 + \left\langle -\sqrt{\alpha}\nabla\mathcal{L}(\boldsymbol{\theta}^k), \sqrt{\alpha} \sum_{m \in \mathcal{M}_c^k} \left( \nabla\mathcal{L}_m(\hat{\boldsymbol{\theta}}_m^k) - \nabla\mathcal{L}_m(\boldsymbol{\theta}^k) \right) \right\rangle. \tag{26}$$

Using $2\mathbf{a}^\top \mathbf{b} = \|\mathbf{a}\|^2 + \|\mathbf{b}\|^2 - \|\mathbf{a} - \mathbf{b}\|^2$, we can re-write the inner product in (26) as

$$\left\langle -\sqrt{\alpha}\nabla\mathcal{L}(\boldsymbol{\theta}^k), \sqrt{\alpha} \sum_{m \in \mathcal{M}_c^k} \left( \nabla\mathcal{L}_m(\hat{\boldsymbol{\theta}}_m^k) - \nabla\mathcal{L}_m(\boldsymbol{\theta}^k) \right) \right\rangle$$

$$= \frac{\alpha}{2} \left\| \nabla\mathcal{L}(\boldsymbol{\theta}^k) \right\|^2 + \frac{\alpha}{2} \left\| \sum_{m \in \mathcal{M}_c^k} \left( \nabla\mathcal{L}_m(\hat{\boldsymbol{\theta}}_m^k) - \nabla\mathcal{L}_m(\boldsymbol{\theta}^k) \right) \right\|^2$$

$$\qquad - \frac{1}{2} \left\| \sqrt{\alpha}\nabla\mathcal{L}(\boldsymbol{\theta}^k) + \sqrt{\alpha} \sum_{m \in \mathcal{M}_c^k} \left( \nabla\mathcal{L}_m(\hat{\boldsymbol{\theta}}_m^k) - \nabla\mathcal{L}_m(\boldsymbol{\theta}^k) \right) \right\|^2$$

$$\overset{(a)}{=} \frac{\alpha}{2} \left\| \nabla\mathcal{L}(\boldsymbol{\theta}^k) \right\|^2 + \frac{\alpha}{2} \left\| \sum_{m \in \mathcal{M}_c^k} \left( \nabla\mathcal{L}_m(\hat{\boldsymbol{\theta}}_m^k) - \nabla\mathcal{L}_m(\boldsymbol{\theta}^k) \right) \right\|^2 - \frac{1}{2\alpha} \left\| \boldsymbol{\theta}^{k+1} - \boldsymbol{\theta}^k \right\|^2 \tag{27}$$

where (a) follows from the LAG update (6).

Combining (26) and (27), and plugging into (25), the claim of Lemma 2 follows.

## B   Missing steps between (9) and (12)

If we choose $\alpha = 1/L$ in Lemmas 1 and 2, it follows that

$$\Delta_{\mathrm{GD}}^k(\boldsymbol{\theta}^k) := -\frac{1}{2L} \left\| \nabla\mathcal{L}(\boldsymbol{\theta}^k) \right\|^2 \tag{28}$$

$$\Delta_{\mathrm{LAG}}^k(\boldsymbol{\theta}^k) := -\frac{1}{2L} \left\| \nabla\mathcal{L}(\boldsymbol{\theta}^k) \right\|^2 + \frac{1}{2L} \left\| \sum_{m \in \mathcal{M}_c^k} \left( \nabla\mathcal{L}_m(\hat{\boldsymbol{\theta}}_m^{k-1}) - \nabla\mathcal{L}_m(\boldsymbol{\theta}^k) \right) \right\|^2. \tag{29}$$

We aim to show that by properly selecting $\alpha$, the following relationship holds

$$\frac{\Delta_{\mathrm{LAG}}^k(\boldsymbol{\theta}^k)}{|\mathcal{M}^k|} \leq \frac{\Delta_{\mathrm{GD}}^k(\boldsymbol{\theta}^k)}{|\mathcal{M}|}. \tag{30}$$

After rearranging terms, (30) is equivalent to

$$\left\| \sum_{m \in \mathcal{M}_c^k} \left( \nabla \mathcal{L}_m(\hat{\boldsymbol{\theta}}_m^{k-1}) - \nabla \mathcal{L}_m(\boldsymbol{\theta}^k) \right) \right\|^2 \leq \frac{|\mathcal{M}_c^k|}{|\mathcal{M}|} \|\nabla \mathcal{L}(\boldsymbol{\theta}^k)\|^2. \tag{31}$$

Note that since we have

$$\left\| \sum_{m \in \mathcal{M}_c^k} \left( \nabla \mathcal{L}_m(\hat{\boldsymbol{\theta}}_m^{k-1}) - \nabla \mathcal{L}_m(\boldsymbol{\theta}^k) \right) \right\|^2 \leq \left| \mathcal{M}_c^k \right| \sum_{m \in \mathcal{M}_c^k} \left\| \nabla \mathcal{L}_m(\hat{\boldsymbol{\theta}}_m^{k-1}) - \nabla \mathcal{L}_m(\boldsymbol{\theta}^k) \right\|^2 \tag{32}$$

if we can further show that

$$\sum_{m \in \mathcal{M}_c^k} \left\| \nabla \mathcal{L}_m(\hat{\boldsymbol{\theta}}_m^{k-1}) - \nabla \mathcal{L}_m(\boldsymbol{\theta}^k) \right\|^2 \leq \frac{\|\nabla \mathcal{L}(\boldsymbol{\theta}^k)\|^2}{|\mathcal{M}|} \tag{33}$$

then we can prove that (30) holds.

However, directly checking (33) at each worker is expensive since i) obtaining $\|\nabla \mathcal{L}(\boldsymbol{\theta}^k)\|^2$ requires information from all the workers; and ii) each worker does not know $\mathcal{M}_c^k$. Hence, we approximate $\|\nabla \mathcal{L}(\boldsymbol{\theta}^k)\|^2$ by

$$\|\nabla \mathcal{L}(\boldsymbol{\theta}^k)\|^2 \approx \frac{1}{\alpha^2} \sum_{d=1}^{D} \xi_d \left\| \boldsymbol{\theta}^{k+1-d} - \boldsymbol{\theta}^{k-d} \right\|^2 \tag{34}$$

and relax $\mathcal{M}_c^k$ to $\mathcal{M}$. Plugging them into (33), we have

$$\left\| \nabla \mathcal{L}_m(\hat{\boldsymbol{\theta}}_m^{k-1}) - \nabla \mathcal{L}_m(\boldsymbol{\theta}^k) \right\|^2 \leq \frac{1}{\alpha^2 M^2} \sum_{d=1}^{D} \xi_d \left\| \boldsymbol{\theta}^{k+1-d} - \boldsymbol{\theta}^{k-d} \right\|^2 \tag{35}$$

which constitute the choice of our trigger condition (12).

## C   Proof of Lemma 3

Using the definition of $\mathbb{V}^k$ in (13), it follows that

$$\mathbb{V}^{k+1} - \mathbb{V}^k = \mathcal{L}(\boldsymbol{\theta}^{k+1}) - \mathcal{L}(\boldsymbol{\theta}^k) + \sum_{d=1}^{D} \beta_d \left\| \boldsymbol{\theta}^{k+2-d} - \boldsymbol{\theta}^{k+1-d} \right\|^2 - \sum_{d=1}^{D} \beta_d \left\| \boldsymbol{\theta}^{k+1-d} - \boldsymbol{\theta}^{k-d} \right\|^2$$

$$\overset{(a)}{\leq} -\frac{\alpha}{2} \left\| \nabla \mathcal{L}(\boldsymbol{\theta}^k) \right\|^2 + \frac{\alpha}{2} \left\| \sum_{m \in \mathcal{M}_c^k} \left( \nabla \mathcal{L}_m(\hat{\boldsymbol{\theta}}_m^k) - \nabla \mathcal{L}_m(\boldsymbol{\theta}^k) \right) \right\|^2 + \sum_{d=2}^{D} \beta_d \left\| \boldsymbol{\theta}^{k+2-d} - \boldsymbol{\theta}^{k+1-d} \right\|^2$$

$$+ \left( \frac{L}{2} - \frac{1}{2\alpha} + \beta_1 \right) \left\| \boldsymbol{\theta}^{k+1} - \boldsymbol{\theta}^k \right\|^2 - \sum_{d=1}^{D} \beta_d \left\| \boldsymbol{\theta}^{k+1-d} - \boldsymbol{\theta}^{k-d} \right\|^2 \tag{36}$$

where (a) uses (8) in Lemma 2.

Decomposing the square distance as

$$\left\| \boldsymbol{\theta}^{k+1} - \boldsymbol{\theta}^k \right\|^2 = \left\| \alpha \nabla \mathcal{L}(\boldsymbol{\theta}^k) + \alpha \sum_{m \in \mathcal{M}_c^k} \left( \nabla \mathcal{L}_m(\hat{\boldsymbol{\theta}}_m^k) - \nabla \mathcal{L}_m(\boldsymbol{\theta}^k) \right) \right\|^2$$

$$\overset{(b)}{\leq} (1+\rho) \alpha^2 \left\| \nabla \mathcal{L}(\boldsymbol{\theta}^k) \right\|^2 + \left( 1 + \rho^{-1} \right) \alpha^2 \left\| \sum_{m \in \mathcal{M}_c^k} \left( \nabla \mathcal{L}_m(\hat{\boldsymbol{\theta}}_m^k) - \nabla \mathcal{L}_m(\boldsymbol{\theta}^k) \right) \right\|^2 \tag{37}$$

where (b) follows from Young's inequality. Plugging (37) into (36), we arrive at (it requires $\frac{L}{2} - \frac{1}{2\alpha} + \beta_1 \geq 0$)

$$\mathbb{V}^{k+1} - \mathbb{V}^k \leq \left( \left( \frac{L}{2} - \frac{1}{2\alpha} + \beta_1 \right) (1+\rho) \alpha^2 - \frac{\alpha}{2} \right) \left\| \nabla \mathcal{L}(\boldsymbol{\theta}^k) \right\|^2$$

$$+ \sum_{d=1}^{D-1} (\beta_{d+1} - \beta_d) \left\| \boldsymbol{\theta}^{k+1-d} - \boldsymbol{\theta}^{k-d} \right\|^2 - \beta_D \left\| \boldsymbol{\theta}^{k+1-D} - \boldsymbol{\theta}^{k-D} \right\|^2$$

$$+ \left( \left( \frac{L}{2} - \frac{1}{2\alpha} + \beta_1 \right) \left( 1 + \rho^{-1} \right) \alpha^2 + \frac{\alpha}{2} \right) \left\| \sum_{m \in \mathcal{M}_c^k} \left( \nabla \mathcal{L}_m(\hat{\boldsymbol{\theta}}_m^k) - \nabla \mathcal{L}_m(\boldsymbol{\theta}^k) \right) \right\|^2. \tag{38}$$

Using $\left(\sum_{n=1}^{N} a_n\right)^2 \leq N \sum_{n=1}^{N} a_n^2$, it follows that

$$\left\| \sum_{m \in \mathcal{M}_c^k} \left( \nabla \mathcal{L}_m(\hat{\boldsymbol{\theta}}_m^k) - \nabla \mathcal{L}_m(\boldsymbol{\theta}^k) \right) \right\|^2 \leq \left| \mathcal{M}_c^k \right| \sum_{m \in \mathcal{M}_c^k} \left\| \nabla \mathcal{L}_m(\hat{\boldsymbol{\theta}}_m^k) - \nabla \mathcal{L}_m(\boldsymbol{\theta}^k) \right\|^2 \tag{39a}$$

$$\overset{(c)}{\leq} \left| \mathcal{M}_c^k \right| \sum_{m \in \mathcal{M}_c^k} L_m^2 \left\| \hat{\boldsymbol{\theta}}_m^k - \boldsymbol{\theta}^k \right\|^2 \tag{39b}$$

$$\overset{(d)}{\leq} \frac{|\mathcal{M}_c^k|^2}{\alpha^2 |\mathcal{M}|^2} \sum_{d=1}^{D} \xi_d \left\| \boldsymbol{\theta}^{k+1-d} - \boldsymbol{\theta}^{k-d} \right\|^2 \tag{39c}$$

where (c) follows the smoothness condition in Assumption 1, and (d) uses the trigger condition (12a) if we derive from (39a) to (39c), uses (12b) if we derive from (39b) to (39c).

Plugging (39) into (38), we have

$$\mathbb{V}^{k+1} - \mathbb{V}^k$$

$$\leq \left( \left( \frac{L}{2} - \frac{1}{2\alpha} + \beta_1 \right)(1+\rho)\alpha^2 - \frac{\alpha}{2} \right) \left\| \nabla \mathcal{L}(\boldsymbol{\theta}^k) \right\|^2$$

$$+ \sum_{d=1}^{D-1} \left( \left( \left( \frac{L}{2} - \frac{1}{2\alpha} + \beta_1 \right)(1+\rho^{-1})\alpha^2 + \frac{\alpha}{2} \right) \frac{\xi_d |\mathcal{M}_c^k|^2}{\alpha^2 |\mathcal{M}|^2} - \beta_d + \beta_{d+1} \right) \left\| \boldsymbol{\theta}^{k+1-d} - \boldsymbol{\theta}^{k-d} \right\|^2$$

$$+ \left( \left( \left( \frac{L}{2} - \frac{1}{2\alpha} + \beta_1 \right)(1+\rho^{-1})\alpha^2 + \frac{\alpha}{2} \right) \frac{\xi_D |\mathcal{M}_c^k|^2}{\alpha^2 |\mathcal{M}|^2} - \beta_D \right) \left\| \boldsymbol{\theta}^{k+1-D} - \boldsymbol{\theta}^{k-D} \right\|^2. \tag{40}$$

After defining some constants to simplify the notation, the proof is then complete.

Furthermore, if the stepsize $\alpha$, parameters $\{\beta_d\}$, and the trigger constants $\{\xi_d\}$ satisfy

$$0 \leq \frac{L}{2} - \frac{1}{2\alpha} + \beta_1; \quad \left( \frac{L}{2} - \frac{1}{2\alpha} + \beta_1 \right)(1+\rho)\alpha^2 - \frac{\alpha}{2} \leq 0 \tag{41a}$$

$$\left( \left( \frac{L}{2} - \frac{1}{2\alpha} + \beta_1 \right)(1+\rho^{-1})\alpha^2 + \frac{\alpha}{2} \right) \frac{\xi_d |\mathcal{M}_c^k|^2}{\alpha^2 |\mathcal{M}|^2} - \beta_d + \beta_{d+1} \leq 0, \forall d = 1, \ldots, D-1 \tag{41b}$$

$$\left( \left( \frac{L}{2} - \frac{1}{2\alpha} + \beta_1 \right)(1+\rho^{-1})\alpha^2 + \frac{\alpha}{2} \right) \frac{\xi_D |\mathcal{M}_c^k|^2}{\alpha^2 |\mathcal{M}|^2} - \beta_D \leq 0 \tag{41c}$$

then Lyapunov function is non-increasing; that is, $\mathbb{V}^{k+1} \leq \mathbb{V}^k$.

**Choice of critical parameters.** We discuss several choices of parameters that satisfy (41).

• If $\beta_1 = \frac{1-\alpha L}{2\alpha}$ so that $\frac{L}{2} - \frac{1}{2\alpha} + \beta_1 = 0$, after rearranging terms, (41) is equivalent to

$$\alpha \leq \frac{1}{L}; \quad \xi_d \leq \frac{2\alpha(\beta_d - \beta_{d+1})|\mathcal{M}|^2}{|\mathcal{M}_c^k|^2}, \forall d \in [1, D-1]; \quad \xi_D \leq \frac{2\alpha\beta_D |\mathcal{M}|^2}{|\mathcal{M}_c^k|^2}. \tag{42}$$

• If $\beta_1 \neq \frac{1-\alpha L}{2\alpha}$, after rearranging terms, (41) is equivalent to

$$\frac{1}{L+2\beta_1} \leq \alpha \leq \frac{1+(1+\rho)^{-1}}{L+2\beta_1}; \tag{43a}$$

$$\xi_d \leq \frac{2\alpha(\beta_d - \beta_{d+1})|\mathcal{M}|^2}{((1+\rho^{-1})(2\alpha\beta_1 + \alpha L - 1) + 1)|\mathcal{M}_c^k|^2}, \quad \forall d = 1, \ldots, D-1 \tag{43b}$$

$$\xi_D \leq \frac{2\alpha\beta_D |\mathcal{M}|^2}{((1+\rho^{-1})(2\alpha\beta_1 + \alpha L - 1) + 1)|\mathcal{M}_c^k|^2}. \tag{43c}$$

i) If $\rho \to 0$ and $\beta_1 \to 0$, (43a) becomes $1/L \leq \alpha \leq 2/L$.
ii) If $\alpha = 1/L$ and $\beta_1 > 0$, (43b) and (43c) reduce to

$$\xi_d \leq \frac{2\alpha(\beta_d - \beta_{d+1})|\mathcal{M}|^2}{(2\alpha\beta_1(1+\rho^{-1}) + 1)|\mathcal{M}_c^k|^2} \quad \text{and} \quad \xi_D \leq \frac{2\alpha\beta_D |\mathcal{M}|^2}{(2\alpha\beta_1(1+\rho^{-1}) + 1)|\mathcal{M}_c^k|^2}. \tag{44}$$

Since (42) is in a simpler form, we will use this choice in the subsequent iteration and communication analysis.

# D Proof of Theorem 1

Using Lemma 3, it follows that (with $\tilde{c}(\alpha, \beta_1) := \frac{L}{2} - \frac{1}{2\alpha} + \beta_1$)

$$
\mathbb{V}^{k+1} - \mathbb{V}^k
$$

$$
\leq - \left( \frac{\alpha}{2} - \tilde{c}(\alpha, \beta_1)\left(1 + \rho\right)\alpha^2 \right) \left\| \nabla \mathcal{L}(\boldsymbol{\theta}^k) \right\|^2
$$

$$
- \left( \beta_D - \left( \tilde{c}(\alpha, \beta_1)\left(1 + \rho^{-1}\right)\alpha^2 + \frac{\alpha}{2} \right) \frac{\xi_D \left| \mathcal{M}_c^k \right|^2}{\alpha^2 |\mathcal{M}|^2} \right) \left\| \boldsymbol{\theta}^{k+1-D} - \boldsymbol{\theta}^{k-D} \right\|^2
$$

$$
- \sum_{d=1}^{D-1} \left( \beta_d - \beta_{d+1} - \left( \tilde{c}(\alpha, \beta_1)\left(1 + \rho^{-1}\right)\alpha^2 + \frac{\alpha}{2} \right) \frac{\xi_d \left| \mathcal{M}_c^k \right|^2}{\alpha^2 |\mathcal{M}|^2} \right) \left\| \boldsymbol{\theta}^{k+1-d} - \boldsymbol{\theta}^{k-d} \right\|^2
$$

$$
\overset{(a)}{\leq} - \left( \alpha\mu - 2\tilde{c}(\alpha, \beta_1)\left(1 + \rho\right)\mu\alpha^2 \right) \left( \mathcal{L}(\boldsymbol{\theta}^k) - \mathcal{L}(\boldsymbol{\theta}^*) \right)
$$

$$
- \left( \beta_D - \left( \tilde{c}(\alpha, \beta_1)\left(1 + \rho^{-1}\right)\alpha^2 + \frac{\alpha}{2} \right) \frac{\xi_D \left| \mathcal{M}_c^k \right|^2}{\alpha^2 |\mathcal{M}|^2} \right) \left\| \boldsymbol{\theta}^{k+1-D} - \boldsymbol{\theta}^{k-D} \right\|^2
$$

$$
- \sum_{d=1}^{D-1} \left( \beta_d - \beta_{d+1} - \left( \tilde{c}(\alpha, \beta_1)\left(1 + \rho^{-1}\right)\alpha^2 + \frac{\alpha}{2} \right) \frac{\xi_d \left| \mathcal{M}_c^k \right|^2}{\alpha^2 |\mathcal{M}|^2} \right) \left\| \boldsymbol{\theta}^{k+1-d} - \boldsymbol{\theta}^{k-d} \right\|^2 \tag{45}
$$

where (a) uses the strong convexity in Assumption 2, implying [9, Appendix B]

$$
2\mu \left( \mathcal{L}(\boldsymbol{\theta}^k) - \mathcal{L}(\boldsymbol{\theta}^*) \right) \leq \left\| \nabla \mathcal{L}(\boldsymbol{\theta}^k) \right\|^2. \tag{46}
$$

With the definition of $c(\alpha; \{\xi_d\})$ as

$$
c(\alpha; \{\xi_d\}) := \min_k \min_{d=1,\dots,D-1} \left\{ \alpha\mu - 2\tilde{c}(\alpha, \beta_1)\left(1 + \rho\right)\mu\alpha^2, 1 - \left( \tilde{c}(\alpha, \beta_1)\left(1 + \rho^{-1}\right)\alpha^2 + \frac{\alpha}{2} \right) \frac{\xi_D \left| \mathcal{M}_c^k \right|^2}{\alpha^2 \beta_D |\mathcal{M}|^2}, \right.
$$

$$
\left. 1 - \frac{\beta_{d+1}}{\beta_d} - \left( \tilde{c}(\alpha, \beta_1)\left(1 + \rho^{-1}\right)\alpha^2 + \frac{\alpha}{2} \right) \frac{\xi_d \left| \mathcal{M}_c^k \right|^2}{\alpha^2 \beta_d |\mathcal{M}|^2} \right\} \tag{47}
$$

from (47), we have

$$
\mathbb{V}^{k+1} - \mathbb{V}^k \overset{(b)}{\leq} - c(\alpha; \{\xi_d\}) \left( \mathcal{L}(\boldsymbol{\theta}^k) - \mathcal{L}(\boldsymbol{\theta}^*) + \sum_{d=1}^{D} \beta_d \left\| \boldsymbol{\theta}^{k+1-d} - \boldsymbol{\theta}^{k-d} \right\|^2 \right)
$$

$$
= - c(\alpha; \{\xi_d\}) \mathbb{V}^k. \tag{48}
$$

Rearranging terms in (45), we can conclude that

$$
\mathbb{V}^{k+1} \leq \left( 1 - c(\alpha; \{\xi_d\}) \right) \mathbb{V}^k. \tag{49}
$$

The $Q$-linear convergence of $\mathbb{V}^k$ implies the $R$-linear convergence of $\mathcal{L}(\boldsymbol{\theta}^k) - \mathcal{L}(\boldsymbol{\theta}^*)$. The proof is complete.

**Iteration complexity.** Since the linear rate constant in (49) is in a complex form, we discuss the iteration complexity under a set of specific parameters (not necessarily optimal). Specifically, we choose

$$
\xi_1 = \dots = \xi_D := \xi < \frac{1}{D} \quad \text{and} \quad \alpha := \frac{1 - D\xi/\eta}{L} \quad \text{and} \quad \beta_d := \frac{(D - d + 1)\xi}{2\alpha\eta}, \ \forall d = 1, \cdots, D \tag{50}
$$

where $\eta$ is a constant. Clearly, (50) satisfies the condition in (42).

Plugging (50) into (47), we have (cf. $\tilde{c}(\alpha, \beta_1) = 0$)

$$
\Gamma := 1 - c(\alpha; \{\xi_d\}) = \max_k \max_{d=1,\dots,D} \left\{ 1 - \frac{1 - D\xi/\eta}{\kappa}, \frac{\eta \left| \mathcal{M}_c^k \right|^2}{|\mathcal{M}|^2}, \frac{D - d + \eta \left| \mathcal{M}_c^k \right|^2 / |\mathcal{M}|^2}{D - d + 1} \right\}. \tag{51}
$$

If we choose $\eta := \sqrt{D\xi}$ such that $\frac{\eta \left| \mathcal{M}_c^k \right|^2}{|\mathcal{M}|^2} < 1$, we can simplify (51) as

$$
\Gamma = \max_k \left\{ 1 - \frac{1 - \sqrt{D\xi}}{\kappa}, \frac{D - 1 + \sqrt{D\xi} \left| \mathcal{M}_c^k \right|^2 / |\mathcal{M}|^2}{D} \right\} \overset{(a)}{=} 1 - \frac{1 - \sqrt{D\xi}}{\kappa}. \tag{52}
$$

where (a) holds since we choose $D \leq \kappa$. With the linear convergence rate in (52), we can derive the iteration complexity as

$$\frac{\mathbb{V}^K}{\mathbb{V}^0} \leq \left(1 - \frac{1 - \sqrt{D\xi}}{\kappa}\right)^K \leq \epsilon$$

$$\Longrightarrow K \log\left(1 - \frac{1 - \sqrt{D\xi}}{\kappa}\right) \leq \log(\epsilon)$$

$$\Longrightarrow \log\left(\frac{1}{\epsilon}\right) \leq K \log\left(1 - \frac{1 - \sqrt{D\xi}}{\kappa}\right)^{-1} \overset{(b)}{\leq} \frac{K}{\frac{\kappa}{1 - \sqrt{D\xi}} - 1}$$

$$\Longrightarrow K \geq \frac{\kappa}{1 - \sqrt{D\xi}} \log\left(\epsilon^{-1}\right) \tag{53}$$

where (b) uses $\log(1 + x) \leq x$, $\forall x > -1$. Therefore, we can conclude that $\mathbb{I}_{\mathrm{LAG}}(\epsilon) = \frac{\kappa}{1 - \sqrt{D\xi}} \log\left(\epsilon^{-1}\right)$.

# E   Proof of Lemma 4

The idea is essentially to show that if (18) holds, then for any iteration $k$, the worker $m$ will not violate the trigger conditions in (12) so that does not communicate with the server at the current iteration, if it has communicated with the server at least once during the previous consecutive $d$ iterations.

Suppose at iteration $k$, the most recent iteration that the worker $m$ did communicate with the server is iteration $k - d'$ with $1 \leq d' \leq d$. Thus, we have $\hat{\boldsymbol{\theta}}_m^{k-1} = \boldsymbol{\theta}^{k-d'}$, which implies that

$$L_m^2 \left\|\hat{\boldsymbol{\theta}}_m^{k-1} - \boldsymbol{\theta}^k\right\|^2 = L_m^2 \left\|\boldsymbol{\theta}^{k-d'} - \boldsymbol{\theta}^k\right\|^2$$

$$= d' L^2 \mathbb{H}^2(m) \sum_{b=1}^{d'} \left\|\boldsymbol{\theta}^{k+1-b} - \boldsymbol{\theta}^{k-b}\right\|^2$$

$$\overset{(a)}{\leq} \frac{\xi_d}{\alpha^2 |\mathcal{M}|^2} \sum_{b=1}^{d'} \left\|\boldsymbol{\theta}^{k+1-b} - \boldsymbol{\theta}^{k-b}\right\|^2$$

$$\overset{(b)}{\leq} \frac{\sum_{b=1}^{d'} \xi_b \left\|\boldsymbol{\theta}^{k+1-b} - \boldsymbol{\theta}^{k-b}\right\|^2}{\alpha^2 |\mathcal{M}|^2} + \frac{\sum_{b=d'+1}^{D} \xi_b \left\|\boldsymbol{\theta}^{k+1-b} - \boldsymbol{\theta}^{k-b}\right\|^2}{\alpha^2 |\mathcal{M}|^2}$$

$$= \text{RHS of (12b)} \tag{54}$$

where (a) follows since the condition (18) is satisfied, so that

$$\mathbb{H}^2(m) \leq \frac{\xi_d}{d\alpha^2 L^2 M^2} \leq \frac{\xi_d}{d' \alpha^2 L^2 M^2} \tag{55}$$

and (b) follows from our choice of $\{\xi_d\}$ such that for $1 \leq d' \leq d$, we have $\xi_d \leq \xi_{d'} \leq \ldots \leq \xi_1$ and $\left\|\boldsymbol{\theta}^{k+1-b} - \boldsymbol{\theta}^{k-b}\right\|^2 \geq 0$. Therefore, the trigger condition (12b) does not activate, and the worker $m$ does not communicate with the server at iteration $k$. With an additional step that $\|\nabla \mathcal{L}_m(\hat{\boldsymbol{\theta}}_m^{k-1}) - \nabla \mathcal{L}_m(\boldsymbol{\theta}^k)\|^2 \leq L_m^2 \|\hat{\boldsymbol{\theta}}_m^{k-1} - \boldsymbol{\theta}^k\|^2$, we can also prove that if $\hat{\boldsymbol{\theta}}_m^{k-1} = \boldsymbol{\theta}^{k-d'}$, the trigger condition (12a) does not activate either.

Note that the above argument holds for any $1 \leq d' \leq d$, and thus if (18) holds, the worker $m$ communicates with the server at most every other $d$ iterations.

# F   Proof of Proposition 5

The condition of communication reduction given in (18) is equivalent to

$$\mathbb{H}^2(m) \leq \frac{\xi_d}{\alpha^2 L^2 |\mathcal{M}|^2 d} := \gamma_d. \tag{56}$$

Together with the definition of heterogeneity score function in (19), given $\gamma_d$, the quantity $h(\gamma_d)$ essentially lower bounds the percentage of workers that communicate with the server at most every other $d$ iterations; that is at most $K/(d+1)$ times until iteration $K$.

To calculate the communication complexity of LAG, we split all the workers into $D + 1$ subgroups:
$\mathcal{M}_0$ - every worker $m$ that does not satisfy $\mathbb{H}^2(m) < \gamma_1$;
$\ldots$
$\mathcal{M}_d$ - every worker $m$ that does satisfy $\mathbb{H}^2(m) < \gamma_d$ but does not satisfy $\mathbb{H}^2(m) < \gamma_{d+1}$;

Figure 6: The area of the light blue polygon lower bounds the quantity $\Delta\bar{\mathbb{C}}(h;\xi)$ in (60). It is generated according to $\gamma_d := 1/(d\gamma_1)$ and $D = 10$.

$\ldots$

$\mathcal{M}_D$ - every worker $m$ that does satisfy $\mathbb{H}^2(m) < \gamma_D$.

The above splitting is according to our claims in Lemma 4, which splits all the workers without overlapping. The neat thing is that for workers in each subgroup $\mathcal{M}_d$, we can upper bound its communication rounds until the current iteration. Hence, the total communication complexity of LAG is upper bounded by

$$
\begin{aligned}
\mathbb{C}_{\text{LAG}}(\epsilon) &= \sum_{m \in \mathcal{M}} \text{Communication rounds of worker } m \\
&= \sum_{d=0}^{D} \text{Total communication rounds of workers in } \mathcal{M}_d \\
&= \sum_{d=0}^{D} |\mathcal{M}_d| \times \frac{\mathbb{I}_{\text{LAG}}(\epsilon)}{d+1} \\
&\overset{(a)}{\leq} \left( 1 - h\left(\gamma_1\right) + \frac{1}{2}\left( h\left(\gamma_1\right) - h\left(\gamma_2\right) \right) + \ldots + \frac{1}{D+1} h\left(\gamma_D\right) \right) M\, \mathbb{I}_{\text{LAG}}(\epsilon) \\
&= \left( 1 - \underbrace{\sum_{d=1}^{D} \left( \frac{1}{d} - \frac{1}{d+1} \right) h\left(\gamma_d\right)}_{\Delta\bar{\mathbb{C}}(h;\{\gamma_d\})} \right) M\, \mathbb{I}_{\text{LAG}}(\epsilon) := \left( 1 - \Delta\bar{\mathbb{C}}(h;\{\gamma_d\}) \right) M\, \mathbb{I}_{\text{LAG}}(\epsilon) \quad (57)
\end{aligned}
$$

where (a) uses the definition of subgroups $\{\mathcal{M}_d\}$ and the function $h(\gamma)$ in (19).

If we choose the parameters as those in (50), we can simplify the expression of (57) and arrive at

$$
\mathbb{C}_{\text{LAG}}(\epsilon) \leq \left( 1 - \Delta\bar{\mathbb{C}}(h;\xi) \right) \frac{M\kappa}{1 - \sqrt{D\xi}} \log(\epsilon^{-1}) \quad (58)
$$

where $\Delta\bar{\mathbb{C}}(h;\{\gamma_d\})$ is written as $\Delta\bar{\mathbb{C}}(h;\xi)$ in this case, because $\gamma_d := \frac{\xi}{(1-\sqrt{D\xi})^2 M^2 d}, \forall d$.

On the other hand, even with a larger stepsize $\alpha = 1/L$, the communication complexity of GD is $\mathbb{C}_{\text{GD}}(\epsilon) := M\kappa \log(\epsilon^{-1})$. Therefore, if we can show that

$$
\frac{1 - \Delta\bar{\mathbb{C}}(h;\xi)}{1 - \sqrt{D\xi}} \leq 1 \quad \Longleftrightarrow \quad \sqrt{D\xi} \leq \Delta\bar{\mathbb{C}}(h;\xi) \quad (59)
$$

then it is safe to conclude that the communication complexity of LAG is lower than that of GD. Using the nondecreasing property of $h$, we have that (cf. the area of the light blue polygon in Figure 6)

$$
\Delta\bar{\mathbb{C}}(h;\xi) \in \left[ \frac{Dh(\gamma_D)}{D+1}, \frac{Dh(\gamma_1)}{D+1} \right] \subseteq \left[ 0, \frac{D}{D+1} \right] \quad (60)
$$

where we use the fact that $0 \leq h(\gamma) \leq 1$. Since for any $\xi \in (0, 1/D)$, there exists a function $h$ such that $\Delta\bar{\mathbb{C}}(h;\xi)$ achieves any value within $[0, D/(D+1)]$. Therefore, we can conclude that if $\xi \leq \frac{D}{(D+1)^2}$ so that $\sqrt{D\xi} \leq D/(D+1)$, there always exists $h(\gamma)$ or a distributed learning setting such that $\mathbb{C}_{\text{LAG}}(\epsilon) < \mathbb{C}_{\text{GD}}(\epsilon)$.

## G  Proof of Theorem 2

Before establishing the convergence in the convex case, we present a critical lemma.

**Lemma 6** *Under Assumptions 1-2, the sequences of Lyapunov functions $\{\mathbb{V}^k\}$ satisfy*

$$\left(\mathbb{V}^k\right)^2 \le \underbrace{\left(\left\|\nabla\mathcal{L}(\boldsymbol{\theta}^k)\right\|^2 + \sum_{d=1}^{D} \beta_d \left\|\boldsymbol{\theta}^{k+1-d} - \boldsymbol{\theta}^{k-d}\right\|^2\right)}_{:= \quad\quad \overline{\mathbb{V}}^k(1)} \underbrace{\left(\left\|\boldsymbol{\theta}^k - \boldsymbol{\theta}^*\right\|^2 + \sum_{d=1}^{D} \beta_d \left\|\boldsymbol{\theta}^{k+1-d} - \boldsymbol{\theta}^{k-d}\right\|^2\right)}_{\times \quad\quad \overline{\mathbb{V}}^k(2)} \tag{61}$$

*where $\overline{\mathbb{V}}^k(1)$ and $\overline{\mathbb{V}}^k(2)$ denote the two terms upper bounding $\left(\mathbb{V}^k\right)^2$, respectively.*

**Proof:** Define two vectors as

$$\mathbf{a}^k := \left[\nabla^\top \mathcal{L}(\boldsymbol{\theta}^k), \sqrt{\beta_1}\left\|\boldsymbol{\theta}^k - \boldsymbol{\theta}^{k-1}\right\|, \ldots, \sqrt{\beta_D}\left\|\boldsymbol{\theta}^{k+1-D} - \boldsymbol{\theta}^{k-D}\right\|\right]^\top \tag{62a}$$

$$\mathbf{b}^k := \left[(\boldsymbol{\theta}^k - \boldsymbol{\theta}^*)^\top, \sqrt{\beta_1}\left\|\boldsymbol{\theta}^k - \boldsymbol{\theta}^{k-1}\right\|, \ldots, \sqrt{\beta_D}\left\|\boldsymbol{\theta}^{k+1-D} - \boldsymbol{\theta}^{k-D}\right\|\right]^\top. \tag{62b}$$

The convexity of $\mathcal{L}(\boldsymbol{\theta})$ implies that

$$\mathcal{L}(\boldsymbol{\theta}^k) - \mathcal{L}(\boldsymbol{\theta}^*) \le \langle\nabla\mathcal{L}(\boldsymbol{\theta}^k), \boldsymbol{\theta}^k - \boldsymbol{\theta}^*\rangle. \tag{63}$$

Recalling the definition of $\mathbb{V}^k$ in (13), it follows that

$$\mathbb{V}^k = \mathcal{L}(\boldsymbol{\theta}^k) - \mathcal{L}(\boldsymbol{\theta}^*) + \sum_{d=1}^{D} \beta_d \left\|\boldsymbol{\theta}^{k+1-d} - \boldsymbol{\theta}^{k-d}\right\|^2$$

$$\le \langle\mathbf{a}^k, \mathbf{b}^k\rangle \le \|\mathbf{a}^k\|\|\mathbf{b}^k\| \tag{64}$$

and squaring both sides of (64) leads to

$$\left(\mathbb{V}^k\right)^2 \le \left(\left\|\nabla\mathcal{L}(\boldsymbol{\theta}^k)\right\|^2 + \sum_{d=1}^{D} \beta_d \left\|\boldsymbol{\theta}^{k+1-d} - \boldsymbol{\theta}^{k-d}\right\|^2\right)\left(\left\|\boldsymbol{\theta}^k - \boldsymbol{\theta}^*\right\|^2 + \sum_{d=1}^{D} \beta_d \left\|\boldsymbol{\theta}^{k+1-d} - \boldsymbol{\theta}^{k-d}\right\|^2\right) \tag{65}$$

from which we can conclude the proof.

Now we are ready to prove Theorem 2. Lemma 3 implies that

$$\mathbb{V}^{k+1} - \mathbb{V}^k \le -\left(\frac{\alpha}{2} - \tilde{c}(\alpha, \beta_1)(1+\rho)\alpha^2\right)\left\|\nabla\mathcal{L}(\boldsymbol{\theta}^k)\right\|^2$$

$$-\left(\beta_D - \left(\tilde{c}(\alpha, \beta_1)(1+\rho^{-1})\alpha^2 + \frac{\alpha}{2}\right)\frac{\xi_D\left|\mathcal{M}_c^k\right|^2}{\alpha^2|\mathcal{M}|^2}\right)\left\|\boldsymbol{\theta}^{k+1-D} - \boldsymbol{\theta}^{k-D}\right\|^2$$

$$-\sum_{d=1}^{D-1}\left(\beta_d - \beta_{d+1} - \left(\tilde{c}(\alpha, \beta_1)(1+\rho^{-1})\alpha^2 + \frac{\alpha}{2}\right)\frac{\xi_d\left|\mathcal{M}_c^k\right|^2}{\alpha^2|\mathcal{M}|^2}\right)\left\|\boldsymbol{\theta}^{k+1-d} - \boldsymbol{\theta}^{k-d}\right\|^2$$

$$\le -c(\alpha; \{\xi_d\})\left(\left\|\nabla\mathcal{L}(\boldsymbol{\theta}^k)\right\|^2 + \sum_{d=1}^{D} \beta_d \left\|\boldsymbol{\theta}^{k+1-d} - \boldsymbol{\theta}^{k-d}\right\|^2\right)$$

$$= -c(\alpha; \{\xi_d\})\overline{\mathbb{V}}^k(1) \tag{66}$$

where the definition of $c(\alpha; \{\xi_d\})$ is given by

$$c(\alpha; \{\xi_d\}) := \min_k \left\{\frac{\alpha}{2} - \tilde{c}(\alpha, \beta_1)(1+\rho)\alpha^2, 1 - \left(\tilde{c}(\alpha, \beta_1)(1+\rho^{-1})\alpha^2 + \frac{\alpha}{2}\right)\frac{\xi_D\left|\mathcal{M}_c^k\right|^2}{\alpha^2\beta_D|\mathcal{M}|^2},\right.$$

$$\left. 1 - \frac{\beta_{d+1}}{\beta_d} - \left(\tilde{c}(\alpha, \beta_1)(1+\rho^{-1})\alpha^2 + \frac{\alpha}{2}\right)\frac{\xi\left|\mathcal{M}_c^k\right|^2}{\alpha^2\beta_d|\mathcal{M}|^2}\right\}. \tag{67}$$

On the other hand, without strong convexity, we can bound $\overline{\mathbb{V}}^k(2)$ as

$$\overline{\mathbb{V}}^k(2) := \left\|\boldsymbol{\theta}^k - \boldsymbol{\theta}^*\right\|^2 + \sum_{d=1}^{D} \beta_d \left\|\boldsymbol{\theta}^{k+1-d} - \boldsymbol{\theta}^{k-d}\right\|^2 \le R \tag{68}$$

where the constant $R$ in the last inequality exists since $\mathcal{L}(\boldsymbol{\theta})$ is coercive in Assumption 2 so that $\mathcal{L}(\boldsymbol{\theta}^*) \le \mathcal{L}(\boldsymbol{\theta}^k) < \infty$ implies $\|\boldsymbol{\theta}^k\| < \infty$ thus $\|\boldsymbol{\theta}^k - \boldsymbol{\theta}^*\| < \infty$ and $\|\boldsymbol{\theta}^k - \boldsymbol{\theta}^{k-1}\| < \infty$.

Plugging (66) and (26) into (61) in Lemma 6, we have

$$\left(\mathbb{V}^k\right)^2 \le \overline{\mathbb{V}}^k(1)\overline{\mathbb{V}}^k(2) \le \frac{R}{c(\alpha; \{\xi_d\})}(\mathbb{V}^k - \mathbb{V}^{k+1}). \tag{69}$$

Using the fact that the non-increasing property of $\mathbb{V}^k$ in Lemma 3, we have that

$$\mathbb{V}^{k+1}\mathbb{V}^k \le \left(\mathbb{V}^k\right)^2 \le \frac{R}{c(\alpha;\{\xi_d\})}(\mathbb{V}^k - \mathbb{V}^{k+1}). \tag{70}$$

Dividing $\mathbb{V}^{k+1}\mathbb{V}^k$ on both sides of (70) and rearranging terms, we have

$$\frac{c(\alpha;\{\xi_d\})}{R} \le \frac{1}{\mathbb{V}^{k+1}} - \frac{1}{\mathbb{V}^k}. \tag{71}$$

Summing up (71), it follows that

$$\frac{Kc(\alpha;\{\xi_d\})}{R} \le \frac{1}{\mathbb{V}^K} - \frac{1}{\mathbb{V}^0} \le \frac{1}{\mathbb{V}^K} \tag{72}$$

from which we can conclude the proof.

# H   Proof of Theorem 3

Lemma 3 implies that

$$
\begin{aligned}
\mathbb{V}^{k+1} - \mathbb{V}^k \le & -\left(\frac{\alpha}{2} - \tilde{c}(\alpha,\beta_1)\left(1+\rho\right)\alpha^2\right)\left\|\nabla\mathcal{L}(\boldsymbol{\theta}^k)\right\|^2 \\
& -\left(\beta_D - \left(\tilde{c}(\alpha,\beta_1)\left(1+\rho^{-1}\right)\alpha^2 + \frac{\alpha}{2}\right)\frac{\xi_D\left|\mathcal{M}_c^k\right|^2}{\alpha^2|\mathcal{M}|^2}\right)\left\|\boldsymbol{\theta}^{k+1-D} - \boldsymbol{\theta}^{k-D}\right\|^2 \\
& -\sum_{d=1}^{D-1}\left(\beta_d - \beta_{d+1} - \left(\tilde{c}(\alpha,\beta_1)\left(1+\rho^{-1}\right)\alpha^2 + \frac{\alpha}{2}\right)\frac{\xi_d\left|\mathcal{M}_c^k\right|^2}{\alpha^2|\mathcal{M}|^2}\right)\left\|\boldsymbol{\theta}^{k+1-d} - \boldsymbol{\theta}^{k-d}\right\|^2 \\
\le & -c(\alpha;\{\xi_d\})\left(\left\|\nabla\mathcal{L}(\boldsymbol{\theta}^k)\right\|^2 + \sum_{d=1}^{D}\beta_d\left\|\boldsymbol{\theta}^{k+1-d} - \boldsymbol{\theta}^{k-d}\right\|^2\right)
\end{aligned}
\tag{73}
$$

Summing up both sides of (73), we have

$$c(\alpha;\{\xi_d\})\sum_{k=1}^{K}\left(\left\|\nabla\mathcal{L}(\boldsymbol{\theta}^k)\right\|^2 + \sum_{d=1}^{D}\beta_d\left\|\boldsymbol{\theta}^{k+1-d} - \boldsymbol{\theta}^{k-d}\right\|^2\right) \le \mathbb{V}^1 - \mathbb{V}^{K+1}. \tag{74}$$

Taking $K \to \infty$, we have that

$$c(\alpha;\{\xi_d\})\lim_{K\to\infty}\sum_{k=1}^{K}\left(\left\|\nabla\mathcal{L}(\boldsymbol{\theta}^k)\right\|^2 + \sum_{d=1}^{D}\beta_d\left\|\boldsymbol{\theta}^{k+1-d} - \boldsymbol{\theta}^{k-d}\right\|^2\right) \le \mathbb{V}^1 \tag{75}$$

where the last inequality holds since the Lyapunov function (13) is lower bounded by $\mathbb{V}^k \ge 0$, $\forall k$, and $\mathbb{V}^1 < \infty$. Given the choice of $\alpha$ and $\{\xi_d\}$ in (41), the constant in (75) is $c(\alpha;\{\xi_d\}) > 0$, and thus two terms in the LHS of (75) are summable, which implies that

$$\sum_{k=1}^{\infty}\left\|\boldsymbol{\theta}^{k+1} - \boldsymbol{\theta}^k\right\|^2 < \infty \tag{76}$$

and likewise that

$$\sum_{k=1}^{\infty}\left\|\nabla\mathcal{L}(\boldsymbol{\theta}^k)\right\|^2 < \infty. \tag{77}$$

Using the implications of summable sequences in [1, Lemma 3], the theorem readily follows.

# I   Communication complexity in (non)convex cases

In the general smooth (possibly nonconvex) case, we compare the communication complexity between LAG and GD, in terms of achieving $\epsilon$-gradient error; e.g., $\min_{k=1,\cdots,K}\left\|\nabla\mathcal{L}(\boldsymbol{\theta}^k)\right\|^2 \le \epsilon$. Denote $\mathbb{C}_{\mathrm{N-GD}}(\epsilon)$ as the communications cost by nonconvex GD to achieve $\epsilon$-gradient error; and $\mathbb{C}_{\mathrm{N-LAG}}(\epsilon)$ denotes the communications cost by LAG in nonconvex cases. In such case, we can establish the following result.

**Proposition 7 (communication complexity in general case)** *Under Assumption 1, with $\gamma_d$, $h(\gamma)$, and $\Delta\bar{\mathbb{C}}(h;\{\gamma_d\})$ defined as in Proposition 5, the communication complexity of LAG is bounded by*

$$\mathbb{C}_{\mathrm{N-LAG}}(\epsilon) \le \left(1 - \Delta\bar{\mathbb{C}}(h;\{\gamma_d\})\right)\frac{\mathbb{C}_{\mathrm{N-GD}}(\epsilon)}{(1 - \sum_{d=1}^{D}\xi_d)}. \tag{78}$$

*Choosing the parameters as* (16), *if the heterogeneity function* $h(\gamma)$ *satisfies that there exists* $\gamma'$ *such that* $\gamma' < \frac{h(\gamma')}{(D+1)DM^2}$, *then we have that*

$$\mathbb{C}_{\text{N-LAG}}(\epsilon) < \mathbb{C}_{\text{N-GD}}(\epsilon). \tag{79}$$

**Proof:** Choosing $\beta_d := \frac{1}{2\alpha} \sum_{\tau=d}^{D} \xi_\tau$ in the Lyapunov function (13), we have

$$\mathbb{V}^k := \mathcal{L}(\boldsymbol{\theta}^k) - \mathcal{L}(\boldsymbol{\theta}^*) + \sum_{d=1}^{D} \frac{(\sum_{j=d}^{D} \xi_j)}{2\alpha} \|\boldsymbol{\theta}^{k+1-d} - \boldsymbol{\theta}^{k-d}\|^2 \tag{80}$$

Using Lemma 2, we arrive at

$$\mathbb{V}^{k+1} - \mathbb{V}^k \leq -\frac{\alpha}{2} \left\| \nabla \mathcal{L}(\boldsymbol{\theta}^k) \right\|^2 + \left( \frac{L}{2} - \frac{1}{2\alpha} + \frac{\sum_{d=1}^{D} \xi_d}{2\alpha} \right) \left\| \boldsymbol{\theta}^{k+1} - \boldsymbol{\theta}^k \right\|^2. \tag{81}$$

If the stepsize is chosen as $\alpha = \frac{1}{L}(1 - \sum_{d=1}^{D} \xi_d)$, we have

$$\mathbb{V}^{k+1} - \mathbb{V}^k \leq -\frac{\alpha}{2} \left\| \nabla \mathcal{L}(\boldsymbol{\theta}^k) \right\|^2. \tag{82}$$

Summing up both sides from $k = 1, \ldots, K$, and initializing $\boldsymbol{\theta}^{1-D} = \cdots = \boldsymbol{\theta}^0 = \boldsymbol{\theta}^1$, we have

$$\sum_{k=1}^{K} \left\| \nabla \mathcal{L}(\boldsymbol{\theta}^k) \right\|^2 \leq \frac{2}{\alpha} \mathbb{V}^1 = \frac{2}{\alpha}(\mathcal{L}(\boldsymbol{\theta}^1) - \mathcal{L}(\boldsymbol{\theta}^*)) = \frac{2L}{1 - \sum_{d=1}^{D} \xi_d}(\mathcal{L}(\boldsymbol{\theta}^1) - \mathcal{L}(\boldsymbol{\theta}^*)) \tag{83}$$

which implies that

$$\min_{k=1,\cdots,K} \left\| \nabla \mathcal{L}(\boldsymbol{\theta}^k) \right\|^2 \leq \frac{2L}{(1 - \sum_{d=1}^{D} \xi_d)K}(\mathcal{L}(\boldsymbol{\theta}^1) - \mathcal{L}(\boldsymbol{\theta}^*)) \tag{84}$$

With regard to GD, it has the following guarantees [33]

$$\min_{k=1,\cdots,K} \left\| \nabla \mathcal{L}(\boldsymbol{\theta}^k) \right\|^2 \leq \frac{2L}{K}(\mathcal{L}(\boldsymbol{\theta}^1) - \mathcal{L}(\boldsymbol{\theta}^*)). \tag{85}$$

Thus, to achieve the same $\epsilon$-gradient error, the iteration of LAG is $(1 - \sum_{d=1}^{D} \xi_d)^{-1}$ times than GD. Similar to the derivations in (57), since the LAG's average communication rounds per iteration is $(1 - \Delta \bar{\mathbb{C}}(h; \{\gamma_d\}))$ times that of GD, we arrive at (78).

If we choose $\xi_1 = \xi_2 = \ldots = \xi_D = \xi$, then $\alpha = \frac{1-D\xi}{L}$, and $\gamma_d = \frac{\xi/d}{\alpha^2 L^2 M^2}$, $d = 1, 2, \ldots, D$. As $h(\cdot)$ is non-decreasing, if $\gamma_D \geq \gamma'$, we have $h(\gamma_D) \geq h(\gamma')$. With the definition of $\Delta \bar{\mathbb{C}}(h; \{\gamma_d\})$ in (20), we can get

$$\Delta \bar{\mathbb{C}}(h; \{\gamma_d\}) = \sum_{d=1}^{D} \left( \frac{1}{d} - \frac{1}{d+1} \right) h(\gamma_d) \geq \sum_{d=1}^{D} \left( \frac{1}{d} - \frac{1}{d+1} \right) h(\gamma_D) \geq \frac{D}{D+1} h(\gamma'). \tag{86}$$

Therefore, the total communications are reduced if

$$\left( 1 - \frac{D}{D+1} h(\gamma') \right) \cdot \frac{1}{1 - D\xi} < 1 \tag{87}$$

which is equivalent to $h(\gamma') > (D+1)\xi$. The condition $\gamma_D \geq \gamma'$ requires

$$\xi/D \geq \gamma'(1 - D\xi)^2 |\mathcal{M}|^2. \tag{88}$$

Obviously, if $\xi > \gamma' D |\mathcal{M}|^2$, then (88) holds. In summary, we need

$$\gamma' < \frac{\xi}{DM^2} < \frac{h(\gamma')}{(D+1)DM^2}. \tag{89}$$

Therefore, we need the function $h$ to satisfy the property: there exists $\gamma'$ such that (89) holds.

## J   Simulation Details

This section provides simulation details of the linear regression and logistic regression tasks.

| Dataset | # features ($d$) | # samples ($N$) | worker index |
|---------|------------------|------------------|--------------|
| Housing | 13 | 506 | 1,2,3 |
| Body fat | 14 | 252 | 4,5,6 |
| Abalone | 8 | 417 | 7,8,9 |

Table 4: A summary of real datasets used in the linear regression tests.

| Dataset | # features ($d$) | # samples ($N$) | worker index |
|---------|------------------|------------------|--------------|
| Ionosphere | 34 | 351 | 1,2,3 |
| Adult fat | 113 | 1605 | 4,5,6 |
| Derm | 34 | 358 | 7,8,9 |

Table 5: A summary of real datasets used in the logistic regression tests.

## J.1 Details for linear regression

For linear regression task, consider the square loss function at worker $m$ as

$$\mathcal{L}_m(\boldsymbol{\theta}) := \sum_{n \in \mathcal{N}_m} \left( y_n - \mathbf{x}_n^\top \boldsymbol{\theta} \right)^2 \tag{90}$$

where $\{\mathbf{x}_n, y_n, \forall n \in \mathcal{N}_m\}$ are data at worker $m$.

**Real datasets.** Performance is tested on the following benchmark datasets [2]; see a summary in Table 4.

• **Housing** dataset [3] contains 506 samples $(\mathbf{x}_n, y_n)$ with $y_n$ representing the median value of house price, which is affected by features in $\mathbf{x}_n$ such as crime rate and weighted distances to Boston employment centers.

• **Body fat** dataset contains 252 samples $(\mathbf{x}_n, y_n)$ with $y_n$ describing the percentage of body fat, which is determined by underwater weighing and various body circumference measurements in $\mathbf{x}_n$.

• **Abalone** dataset contains 417 samples $(\mathbf{x}_n, y_n)$ with $y_n$ for the age of abalone and $\mathbf{x}_n$ for the physical measurements of abalone, e.g., sex, height, and shell weight.

## J.2 Details for logistic regression

For logistic regression, consider the binary logistic regression problem

$$\mathcal{L}_m(\boldsymbol{\theta}) := \sum_{n \in \mathcal{N}_m} \log \left( 1 + \exp(-y_n \mathbf{x}_n^\top \boldsymbol{\theta}) \right) + \frac{\lambda}{2} \|\boldsymbol{\theta}\|^2. \tag{91}$$

where $\lambda = 10^{-3}$ is the regularization constant.

**Real datasets.** Performance is tested on the following real datasets; see a summary in Table 5.

• **Ionosphere** dataset [4] is to predict whether it is a "good" radar return or not – it is "good" if the features in $\mathbf{x}_n$ show evidence of some type of structure in the ionosphere.

• **Adult** dataset [5] contains samples that predict whether a person makes over $50K$ a year based on features in $\mathbf{x}_n$ such as work-class, education, and marital-status.

• **Derm** dataset [6] is for differential diagnosis of erythemato-squaxous diseases, which is determined by clinical and histopathological attributes in $\mathbf{x}_n$ such as erythema, family history, focal hypergranulosis and melanin incontinence.

**Synthetic datasets** for both linear and logistic regression tasks are generated as follows. For each worker, we generate 50 samples $\mathbf{x}_n \in \mathbb{R}^{50}$ from the standard Gaussian distribution, and rescale the data to mimic the increasing smoothness constants $L_1, \cdots, L_M$ as

$$[L_1, \ldots, L_9] = [4, 6.76, 12.67, 25.97, 57.06, 131.92, 316.03, 775.26, 1931.57]. \tag{92}$$

and for the uniform smoothness constants $L_1 = L_M = 4$.

Due to the limited space in the manuscript, this section will present the parameters used in the logistic regression test on the Gisette dataset. The task is a binary classification which discriminates between to confusable handwritten digits: the four and the nine. The dataset was taken from [7] which was constructed from the MNIST data [8]. After random selecting subset of samples and eliminating all-zero features, it contains 2000 samples $\mathbf{x}_n \in \mathbb{R}^{4837}$. We randomly split this dataset into nine workers.

As those in Section 4, for LAG-WK, we choose $\xi_d = \xi = 1/D$ with $D = 10$, and for LAG-PS, we choose more aggressive $\xi_d = \xi = 10/D$ with $D = 10$ in all the tests. Stepsizes for LAG-WK, LAG-PS, and GD

are chosen as $\alpha = 1/L$; to optimize performance and guarantee empirical stability, stepsizes for Cyc-IAG and Num-IAG are $\alpha = 1/(ML)$. The regularization parameter is set to $\lambda = 10^{-4}$.