[Reviews · NeurIPS 2018]

Reviewer 1



This paper explores the question of minimizing the communication among workers while solving an optimization problem in a distributed fashion. In particular, the authors argue that most of the existing work in this direction focused on minimizing the amount of data during each message exchange between the works. In contrast, the authors focus on reducing the number of such exchanges (or communication rounds) during the optimization procedures. The authors argue that reducing the number of rounds is more beneficial as it does not degrade the rate of convergence for a wide class of objective functions. The authors propose a simple approach to reduce the communication: a worker sends the current values of the gradient (based on its data) only if this gradient is significantly far from the previous gradient supplied by this worker, leading to the name lazily aggregated gradient (LAG). The authors propose two variants of this approach depending on whether the parameter server or the individual worker decides when to send the current estimate of the gradient. Under natural assumptions of the objective functions, such as smoothness, convexity, or strong-convexity, the authors show that their approach results into the convergence rate similar to that of the conventional gradient descent method where each worker sends the current estimate of the gradient (based on its data) during each iteration. Using a detailed experimental evaluation, the authors show that the LAG approach significantly outperforms some of the previous approaches to reduce the communication in terms of both the convergence rate and the total amount of communication among the workers. The paper proposes a simple yet effective method to reduce the overall communication load for a distributed optimization problem. The proposed method has favorable performance guarantees and has been shown work well with both synthetic and real datasets. The authors argue that the proposed solution can be combined with other approaches to reduce communication, such as quantized gradients and second-order methods. Can authors comment on how the theoretical guarantees might be affected by bringing in these additional ideas?

Reviewer 2



This paper proposed an implementation of gradient descent by using stale gradient computation on some data. I found this method is useful and practical. This paper is well written and easy to understand. My only concern is the connection between this paper and some recent works, e.g., [A] and [B], that study the convergence and speedup of asynchronous parallel SGD. To me, the proposed method and the followed analysis are just a special case of asynchronous SGD with data variance equal to zero (\sigma = 0). Authors are expected to highlight the novelty of the propped method over [A] and [B]. [A] Asynchronous stochastic gradient decent for non convex optimization [B] A comprehensive linear speedup analysis for asynchronous stochastic parallel optimization from zeroth-order to first-order ================= Authors mostly addressed my concerns. Authors are also expected to compare / discuss the connection to SAG algorithm which is very similar to this paper except assuming randomized sampling.

Reviewer 3



Consider synchronous non-stochastic gradient descent where data is distributed across M workers and each worker computes the gradient at each time step and sends it to the server. Here at each round, the server needs to send the model to all workers and all workers need to send their updates to the server. Authors propose lazy aggregated gradient for distributed learning, where at each round only a fraction of them send their updates to a server. They call this lazily aggregated descent and show that it’s optimization performance empirically is similar to that of the full gradient descent. Further, they show that the theoretical performance is also similar for strongly convex, convex, and smooth non-convex functions. The results are interesting and I recommend acceptance. I have few high level questions and minor comments: 1. Gradient descent results can be improved using accelerated methods (e.g., to sqrt{condition number}). Do similar results hold for LAG? 2. Is this method related to methods such as (SAG / SAGA / SVRG), for example see Algorithm 1 in https://arxiv.org/pdf/1309.2388.pdf, where they choose \mathcal{M} randomly. 3. It might be interesting to see the performance of these models on neural networks. 4. What is capital D in Equation 11? 5. In line 133, alpha is set to 1/L. But wouldn’t any alpha < 1/L also works? It is not clear to me how to find the exact smoothness constant.